# EqNIO: Subequivariant Neural Inertial Odometry

**Royina Karegoudra Jayanth,**[*] **Yinshuang Xu,**[*] **Ziyun Wang, Evangelos Chatzipantazis,**
**Kostas Daniilidis**[†]**, Daniel Gehrig**[†]
University of Pennsylvania
{royinakj,xuyin,ziyunw,vaghat,dgehrig}@seas.upenn.edu
kostas@cis.upenn.edu

## Abstract

Neural network-based odometry using accelerometer and gyroscope readings from a single IMU can achieve robust, and low-drift localization capabilities, through the use of *neural displacement priors (NDPs)*. These priors learn to produce denoised displacement measurements but need to ignore data variations due to specific IMU mount orientation and motion directions, hindering generalization. This work introduces EqNIO, which addresses this challenge with *canonical displacement priors*, i.e., priors that are invariant to the orientation of the gravity-aligned frame in which the IMU data is expressed. We train such priors on IMU measurements, that are mapped into a learnable canonical frame, which is uniquely defined via three axes: the first is gravity, making the frame gravity aligned, while the second and third are predicted from IMU data. The outputs (displacement and covariance) are mapped back to the original gravity-aligned frame. To maximize generalization, we find that these learnable frames must transform equivariantly with global gravity-preserving roto-reflections from the subgroup $O_g(3) \subset O(3)$, acting on the trajectory, rendering the NDP $O(3)$-*subequivariant*. We tailor specific linear, convolutional, and non-linear layers that commute with the actions of the group. Moreover, we introduce a bijective decomposition of angular rates into vectors that transform similarly to accelerations, allowing us to leverage both measurement types. Natively, angular rates would need to be inverted upon reflection, unlike acceleration, which hinders their joint processing. We highlight EqNIO's flexibility and generalization capabilities by applying it to both filter-based (TLIO), and end-to-end (RONIN) architectures, and outperforming existing methods that use *soft* equivariance from auxiliary losses or data augmentation on various datasets. We believe this work paves the way for low-drift and generalizable neural inertial odometry on edge devices. The project details and code can be found at https://github.com/RoyinaJayanth/EqNIO.

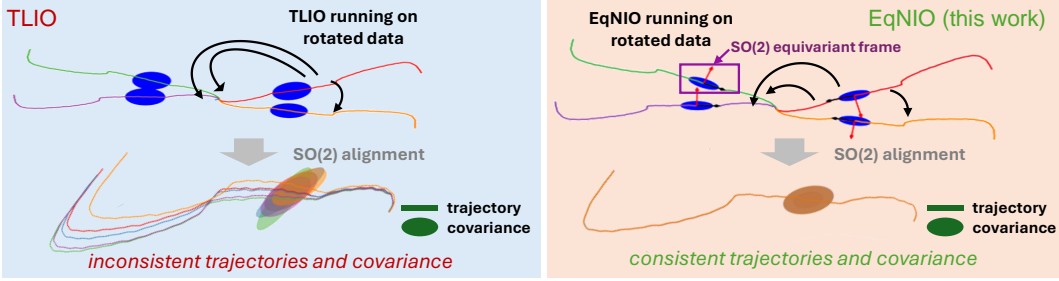

Figure 1: Neural Displacement Priors (NDPs) that rely on data augmentation (TLIO, left) produce a different trajectory for different reference frames while strictly equivariant approaches (EqNIO, right) yield one trajectory independent of the reference frame. EqNIO achieves this by learning an equivariant canonical frame aligned, per definition, with the predicted covariance (ellipsoids,right). In test time, the gravity-aligned IMU orientation estimate of the Kalman filter is the reference frame.

---

[*]denotes equal contribution, † denotes equal advising

# 1 INTRODUCTION

Inertial Measurement Units (IMUs) are commodity sensors that measure the accelerations and angular velocities of a body. Due to their low cost, they are widely used in diverse applications such as robot navigation and Mixed Reality for precise and rapid tracking of body frames. However, since they are differential sensors, relying only on IMUs invariably results in drift. Traditional Visual Inertial Odometry (VIO) approaches can effectively mitigate this drift by combining IMU measurements with features extracted from camera images. Still, these images are of limited use in high-speed scenarios with challenging lighting conditions, since they can suffer from saturation, and blurring artifacts.

Recently, a novel class of methods has emerged that instead mitigates this drift with neural displacement priors (NDPs), that are directly learned from IMU data alone (Liu et al., 2020; Herath et al., 2020). These methods perform competitively with VIO methods despite using only a single IMU sensor. However, learning generalizable priors proves challenging: while identifying specific motion patterns, they must learn to ignore data differences due to particular IMU mount orientations, and the direction of the motion patterns. However, they often fail to do so in practice (see Fig. 1, left state-of-the-art method TLIO (Liu et al., 2020)), yielding large trajectory variations when observing simply rotated input data. These variations persist, in spite of data augmentation strategies (TLIO Liu et al. (2020), RONIN Herath et al. (2020)) that include random data rotations or the enforcement of equivariance constraints via auxiliary consistency losses during training (RIO Cao et al. (2022)).

In this work, we simplify this task by introducing **EqNIO**, which leverages *canonical displacement priors (CDPs)*. CDPs are invariant to IMU orientation and, as a result, easier to learn, more robust, and more generalizable than regular NDPs. EqNIO first maps the IMU data into a learnable, equivariant, and gravity-aligned frame $F$, before passing them to the CDP, and mapping the outputs back to the original frame. Our design can flexibly integrate arbitrary off-the-shelf methods such as TLIO (Liu et al., 2020) and RONIN (Herath et al., 2020), provided that suitable change of basis maps are defined for the network outputs (displacement and covariance of TLIO, and only linear velocity for RONIN). The frame $F$ is composed of two transformations: The first, gravity-alignment, aligns the $z$-axis with the gravity direction, estimated from an off-the-shelf filter (Liu et al., 2020), and is traditionally used in inertial odometry. The second is a learnable, gravity-preserving roto-reflection.

To be maximally generalizable, we show that this roto-reflection must generalize to arbitrary gravity-preserving roto-reflections of the input data. We design a specific model that achieves this by processing IMU data *equivariantly*, *i.e.*, in a way that commutes with the action of gravity-preserving roto-reflections on the input data. We identify these roto-reflections as elements of the group $O_g(3)$, a subgroup of the orthogonal group $O(3)$, which is isomorphic to $O(2)$. Equivariance is ensured by a preliminary, unique preprocessing step that maps accelerometer and gyroscope measurements into a space that transforms consistently under the group action, and subsequent processing with equivariant MLPs, convolutions, and non-linear layers. Due to the isomorphism with $O(2)$ these layers are $O(2)$ equivariant, and, due to the subgroup property, also called $O(3)$ *subequivariant*.

**Contributions:** *(i)* We apply a canonicalization scheme that maps IMU measurement, and NDP outputs to and from a gravity-aligned, subequivariant, canonical frame. This procedure can be flexibly applied to arbitrary off-the-shelf network architectures. NDPs trained on such canonical data produce inherently more robust, and generalizable results than previous work. *(ii)* We formalize the group actions of gravity-preserving roto-reflection from $O_g(3)$ on IMU measurements and derive unique preprocessing steps that map both accelerometer and gyroscope measurements into a space in which these actions are consistent. *(iii)* We tailor an $O(2)$ equivariant network that regresses canonical roto-reflections from 2D vector features, and 1D scalar derived from IMU data. It leverages specialized $O(2)$ equivariant MLPs and convolution, to process vector features, conventional layers to process scalar features, and equivariant non-linearities to mix vector and scalar features.

We demonstrate the generality of our framework by applying it to two neural inertial odometry methods, TLIO (Liu et al., 2020), and RONIN (Herath et al., 2020). Extensive qualitative and quantitative results comparing EqNIO against previous works across diverse benchmarks establish a new state-of-the-art in inertial-only odometry. EqNIO significantly enhances the accuracy, reliability, and generalization of existing methods.

# 2 RELATED WORK

Neural inertial odometry is highly related to inertial navigation systems (INS) which can be broad classified into estimation-based, multi-sensor data fusion-based (MSDF) techniques, and learning-

based techniques. Recent works in inertial attitude estimation (Asgharpoor Golroudbari & Sabour, 2023; Ge et al., 2024), VIO (van Goor & Mahony, 2023), LIDAR odometry (Zheng et al., 2024) and SLAM (Kim & Sukkarieh, 2003) are highly related and are hence discussed in App. A.1.

**Model-based Inertial Odometry:** Purely Inertial Odometry can be broadly classified into two categories: kinematics-based and learning-based approaches. Kinematics-based approaches (Leishman et al., 2014; Titterton et al., 2004; Bortz, 1971) leverage analytical solutions based on double integration that suffers from drift accumulation over time when applied to consumer-grade IMUs. To mitigate this drift, loop closures (Solin et al., 2018), and other pseudo measurements derived from IMU data that are drift-free have been explored (Groves, 2015; Hartley et al., 2020; Brajdic & Harle, 2013). In the context of Pedestrian Dead Reckoning (Jimenez et al., 2009), these include step counting (Ho et al., 2016; Brajdic & Harle, 2013), detection of the system being static (Foxlin, 2005; Rajagopal, 2008) and gait estimation (Beaufils et al., 2019).

**Learning-based Inertial Odometry:** Recently, RIDI (Yan et al., 2018), PDRNet (Asraf et al., 2022) and RONIN (Herath et al., 2020) proposed CNN, RNN, and TCN-based velocity regression. RIDI uses these velocities to correct the IMU measurements, while RONIN directly integrates them while assuming given orientation information. Denoising networks either regress IMU biases (Brossard et al., 2020b; Buchanan et al., 2023; Brossard et al., 2020a) denoised IMU measurements directly (Steinbrener et al., 2022). While Buchanan et al. (2023) uses constant covariance, AI-IMU (Brossard et al., 2020a) estimates the covariance for automotive applications. Displacement-based methods like IONet (Chen et al., 2018a), TLIO (Liu et al., 2020), RNIN-VIO (Chen et al., 2021a), and IDOL (Sun et al., 2021) directly estimate 2D/3D displacement. Unlike TLIO which regresses a diagonal covariance matrix, Russell & Reale (2021) estimates the full covariance matrix parameterized via Pearson correlations. RNIN-VIO extends TLIO to continuous human motion adding a loss function for long-term accuracy. Unlike these methods, EqNIO learns canonical displacement priors and thus generalizes better to arbitrary IMU orientation and motion directions.

**Equivariant Inertial Odometry:** The previous learning-based approaches (Liu et al., 2020; Chen et al., 2021a) use $SO(2)$ augmentation strategies to achieve approximate $SO(2)$ equivariance. MotionTransformer (Chen et al., 2019) used GAN-based RNN encoder to transfer IMU data into domain-invariant space by separating the domain-related constant. Recently, RIO (Cao et al., 2022) demonstrated the benefits of approximate $SO(2)$ equivariance with an auxiliary loss, introduced Adaptive Test Time Training (TTT), and uncertainty estimation via ensemble of models (See Appendix A.4 for more details). We propose integrating strict equivariance by design directly into the framework. Additionally, no prior work has addressed reflection equivariance, which requires specific preprocessing of gyroscope data for it to adhere to the right-hand rule. Our novel $O(2)$ equivariant framework can be seamlessly integrated with existing learning-based inertial navigation systems.

**Equivariant Networks:** Group equivariant networks (Cohen & Welling, 2016) commute by design with group actions on the input, and have been tailored to a variety of inputs and architecture designs. These include point clouds (Thomas et al., 2018; Chen et al., 2021b; Deng et al., 2021), 2D (Worrall et al., 2017; Weiler & Cesa, 2019), 3D (Weiler et al., 2018; Esteves et al., 2019), and spherical images (Cohen et al., 2018; Esteves et al., 2018; 2020; 2023), graphs (Satorras et al., 2021) and general manifolds (Cohen et al., 2019b;a; Weiler et al., 2021; Xu et al., 2024). Yet, equivariant networks tailored to IMU data and their symmetries have not been studied, and are introduced in the current work. Our method draws on general theories and methods developed for equivariance to $E(n)$ its subgroups. Cesa et al. (2021); Xu et al. (2022) use Fourier analysis to design steerable CNN kernels on homogeneous space, while Finzi et al. (2021) proposed an algorithm for finding a kernel by solving a linear equivariant map constraint. Villar et al. (2021) demonstrated that any $O(n)$ equivariant function can be represented using a set of scalars and vectors. However, applying these to the neural integration of IMUs is not straightforward as gravity's presence introduces subequivariance, angular velocity in the input data follows the right-hand rule, and the input is a sequence with a time dimension. Related approaches (Han et al., 2022; Chen et al., 2023) tackle subequivariance using equivariant graph networks and calculating gram matrices achieving simple $O(2)$ equivariance. However, dealing with data that obey the right-hand rule (*e.g.* angular rates), has been underexplored, and is addressed in the current work. Canonicalization is a prevalent approach to equivariance. Kaba et al. (2023) first introduced learned canonicalization functions for equivariant networks, which we adopt in our design. We discuss additional works on equivariant canonicalization in Appendix A.1.

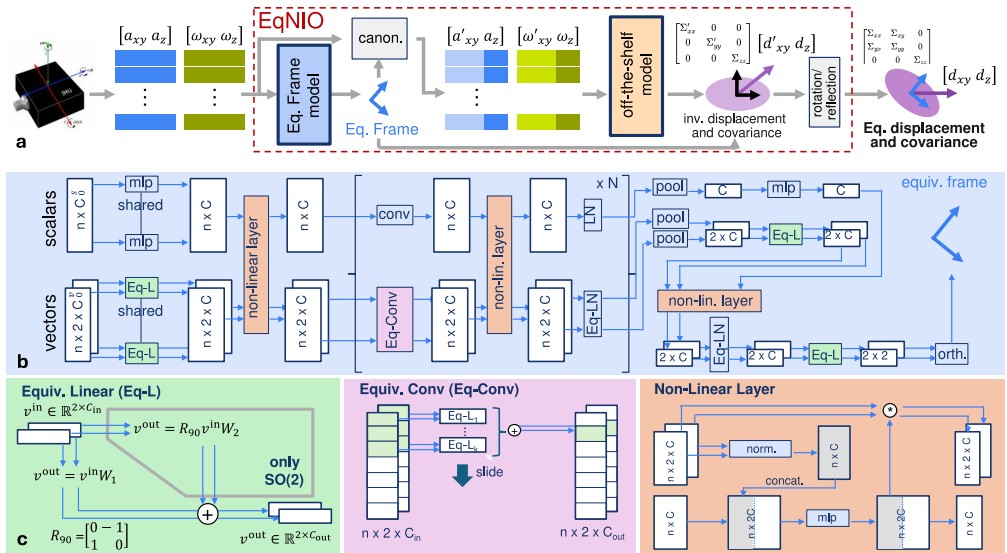

Figure 2: EqNIO (a) processes gravity-aligned IMU measurements, $\{(a_i, \omega_i)\}_{i=1}^n$. An equivariant network (blue) predicts a canonical equivariant frame $F$ into which IMU measurements are mapped, *i.e. canonicalized*, yielding invariant inputs $\{(a_i', \omega_i')\}_{i=1}^n$. A conventional neural network then predicts invariant displacement ($d'$) and covariance ($\Sigma'$) which are mapped back yielding equivariant displacement ($d$) and covariance ($\Sigma$). The equivariant network (b) takes as input $n \times C_0^s$ scalars, and $n \times C_0^v$ vectors: Vectors are processed by equivariant layers (Eq-L, Eq-Conv, Eq-LN), while scalars are separately processed with conventional layers. Eq-L (green) uses two weights $W_1, W_2$ for SO(2) equivariance, and only $W_1$ for O(2) equivariance. Eq-Conv (pink) uses Eq-L to perform 1-D convolutions over time. The equivariant non-linear layer (orange) mixes vector and scalar features.

## 3 PROBLEM SETUP

This paper targets neural inertial odometry using data from a single IMU, comprised of an accelerometer (giving linear accelerations $a_i \in \mathbb{R}^3$) and gyroscope (giving angular velocity $\omega_i \in \mathbb{R}^3$). IMU's measure sequences of data $\{(a_i, \omega_i)\}_{i=1}^n$, each expressed in the local IMU body frame, $b$, at time $t_i$. These are related to the true IMU acceleration $\bar{a}_i$ and angular rates $\bar{\omega}_i$ via

$$\tilde{\omega}_i = \bar{\omega}_i + b_i^g + \eta_i^g \qquad \tilde{a}_i = \bar{a}_i - {}_b^w R_i^T g + b_i^a + \eta_i^a \tag{1}$$

where $g$ is gravity pointing downward in world frame $w$, ${}_b^w R_i$ is the transformation between $b$ and $w$ at time $t_i$, and $b_i^g, b_i^a$ and $\eta_i^g, \eta_i^a$ are IMU biases and noises respectively. Naively integrating angular rates and accelerations to get positions $p_i$ and orientations ${}_b^w R_i$ leads to significant drift due to sensor noise and unknown biases. We thus turn our attention to neural displacement priors $\Phi$, which regress accurate 2D linear velocities (Herath et al., 2020) or 3D displacement $d \in \mathbb{R}^3$ and covariances $\Sigma \in \mathbb{R}^{3\times3}$ (Liu et al., 2020) from sequences of bias-corrected and gravity aligned IMU measurements

$$\omega_i = {}_b^g R_i(\tilde{\omega}_i - b^g) \qquad a_i = {}_b^g R_i(\tilde{a}_i - b^a) \tag{2}$$

where ${}_b^g R_i$ aligns the z-axis with gravity, and is defined as ${}_b^g R_i = R_\gamma {}_b^w R_i$ with some unobservable yaw rotation $R_\gamma$. The neural displacement prior has the form

$$d, \Sigma = \Phi\left(\{(a_i, \omega_i)\}_{i=1}^n\right) \tag{3}$$

where $d \in \mathbb{R}^3$ denotes displacement on the time interval $[t_1, t_n]$, and $\Sigma \in \mathbb{R}^{3\times3}$ denotes associated covariance prediction. For instance, Liu et al. (2020), uses these network predictions as measurements and fuses them in an EKF estimating the IMU state in $w$, *i.e.* orientation, position, velocity, and IMU biases. Preliminaries on terms used in inertial odometry are included in App. A.2.2 and details of EKF and IMU measurement model are included in App. A.6.

We simplify the learning of informative priors by suitably canonicalizing the IMU measurements in two steps: First, we gravity-align IMU measurements by rotating them into the frame ${}_b^g R$, such that the z-axis of the IMU frame and world frame coincide. We use the EKF orientation state from (Liu et al., 2020) to find the gravity direction (see App. A.6). Later we empirically show the robustness of our method to noise originating from this estimation. In what follows we thus

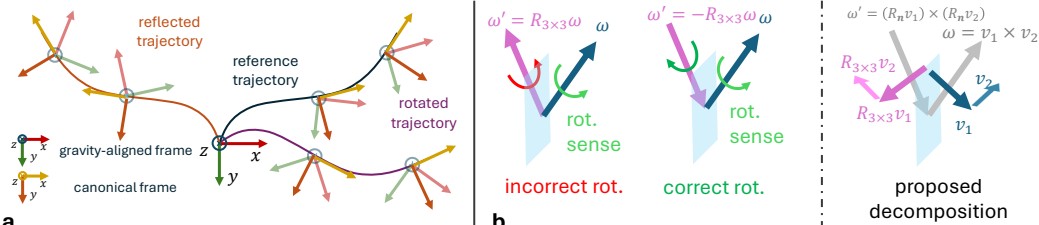

Figure 3: Symmetries in neural inertial odometry. (a) An IMU undergoes three trajectories in $xy$-plane, each related to a reference (blue) via rotation (purple) and/or reflection (orange) around gravity (parallel to the $z$-axis). At a fixed time, IMU measurements on different trajectories, expressed in the corresponding local gravity-aligned frame (red-green) differ only by an unknown yaw roto-reflection $R_{3\times3}$. Mapping these measurements to a canonical frame (yellow-red) that transforms equivariantly under roto-reflections of the trajectory eliminates this ambiguity enhancing the sample efficiency of downstream neural networks. (b) Expressed in alternative roto-reflected frames, acceleration, and angular rates transform as $a' = R_{3\times3}a$ and $\omega' = \det(R_{3\times3})R_{3\times3}\omega$. Angular rate must follow the right-hand-rule, and thus be also inverted when reflected. To ensure a similar transformation rule as $a$, we decompose $\omega = v_1 \times v_2$ and process $v_1, v_2$ instead, which transform as $v'_{1/2} = R_{3\times3}v_{1/2}$

assume accelerations and angular rates to be expressed in the gravity-aligned frame and illustrate these frames in Fig. 3 (a) for three rotated trajectories. Gravity alignment reduces data variability by two degrees of freedom. However, this frame is not unique, since simply rotating it around z or reflecting it across planes parallel to z (applications of rotations or roto-reflections from the groups $SO_g(3) = \{R \in SO(3)|Rg = g\}$ and $O_g(3) = \{R \in O(3)|Rg = g\}$) result in new valid gravity-aligned frames. Hence, secondly, we predict a canonical frame and map the IMU data into this frame, which we later show to be subequivariant to roto-reflections. In what follows we will restrict our discussion to the $O_g(3)$ case but note that, where not explicitly stated, this discussion carries over to $SO_g(3)$ as well. Next, we will introduce our canonicalization procedure to ensure better network generalization.

## 4 METHODOLOGY

Our goal is to predict a canonical yaw frame $F = \Psi(\{(a_i, \omega_i)\}_{i=1}^n) \in O_g(3)$ from data, which generalizes across arbitrary yaw orientations. We use this frame to map IMU data into a canonical frame before giving as input to the NN, and mapping the outputs back, (see Fig. 2 (a)),*i.e.*

$$d', \Sigma' = \Phi(\{(a'_i, \omega'_i)\}_{i=1}^n) \qquad \text{with } a'_i = \rho_a(F^{-1})a_i \text{ and } \omega'_i = \rho_\omega(F^{-1})\omega_i, \tag{4}$$

where $a', \omega'$ are expressed in the canonical frame. Finally, we map $d', \Sigma'$ back to the original frame via $d = \rho_d(F)d'$ and $\text{vec}(\Sigma) = \rho_\Sigma(F)\text{vec}(\Sigma')$. Here vec(.) stacks the columns of $\Sigma$ into a single vector, and $\rho$ is a homomorphism that maps group elements $F$ to corresponding matrices, called *matrix representations*. These capture the transformation of $a, \omega, d$ and $\Sigma$ under the action of $F$.

While $\rho_a(F) = \rho_d(F) = F_{3\times3}$, with $F_{3\times3} \in \mathbb{R}^{3\times3}$ being the rotation matrix corresponding to element $F$, covariances transform as $\rho_\Sigma(F) = F_{3\times3} \otimes F_{3\times3}$, where $\otimes$ is the Kronecker product. Unfortunately, reflections ($\det(F_{3\times3}) = -1$) induce a reflection *and inversion* of angular rates to preserve the right-hand-rule (see Fig. 3 b), *i.e.* $\rho_\omega(F) = \det(F_{3\times3})F_{3\times3}$, and for reflections. As discussed later $\rho_\omega(F) \neq \rho_a(F)$ hinders joint processing of accelerations and angular rates. Next, we will discuss the design of $\Psi$ which ensures generalization across arbitrary yaw rotations.

### 4.1 EQUIVARIANT FRAME

Here we derive a property of the frame network $\Psi$ such that it can generalize to arbitrary roto-reflections of the IMU body frame. To generalize, canonical IMU measurement inputs $a'_i, \omega'_i$ to the network must look identical under arbitrary roto-reflections $R \in O_g(3)$. Let $a_i, \omega_i$, and $a^*_i, \omega^*_i$ denote quantities before and after application of $R$. Then $a^*_i = \rho_a(R)a_i$ and $\omega^*_i = \rho_\omega(R)\omega_i$. Enforcing identical inputs under both rotations, *i.e.* $a^{*'}_i = a'_i$ we have

$$a^{*'}_i = \rho_a(F^{*-1})a^*_i \quad \omega^{*'}_i = \rho_\omega(F^{*-1})\omega^*_i \tag{5}$$

We see that choosing $F^* = RF$, *i.e.* that $F$ transforms equivariantly leads to

$$a^{*'}_i = \rho_a(F^{*-1})a^*_i = \rho_a(F^{-1}R^{-1})\rho_a(R)a_i = \rho_a(F^{-1}R^{-1}R)a_i = a^*_i \tag{6}$$

where we have used the fact that $\rho_a$ is a homomorphism. This shows the invariance of $a_i'$ and a similar proof can be done for $\omega_i'$. This equality puts a constraint on the NN that estimates $F$, namely

$$RF = \Psi(\{(\rho_a(R)a_i, \rho_\omega(R)\omega_i)\}_{i=1}^n) \tag{7}$$

*i.e.* $\Psi$ must be a function that is *equivariant* with respect to group actions by elements from $O_g(3)$. Since this is a subgroup of $O(3)$ we also say that $\Psi$ must be *subequivariant* with respect to $O(3)$.

In addition, this equivariance property of $\Psi$ induces end-to-end equivariance to predicted displacements $d = \rho_d(F)d'$ and covariances $\text{vec}(\Sigma) = \rho_\Sigma(F)\text{vec}(\Sigma')$. This is because

$$d^* = \rho_d(F^*)d^{*\prime} = \rho_d(RF)d' = \rho_d(R)\rho_d(F)d' = \rho_d(R)d \tag{8}$$

$$\text{vec}(\Sigma^*) = \rho_\Sigma(F^*)\text{vec}(\Sigma'^*) = \rho_\Sigma(RF)\text{vec}(\Sigma') = \rho_\Sigma(R)\rho_\Sigma(F)\text{vec}(\Sigma') = \rho_\Sigma(R)\text{vec}(\Sigma) \tag{9}$$

using, again, the homomorphism of $\rho$ and the fact that $d'$, $\text{vec}(\Sigma')$ are, by construction of equation 6, invariant to rotations by $R$.

**Diagonal Covariance:** We show empirically in Sec. 6 that the diagonal parameterization of $\Sigma'$ aids in stabilization and convergence of the network. Therefore, we assume the displacement uncertainties $\Sigma_{d,xz} = \Sigma_{d,yz} = 0$ and without loss in generality choose $\Sigma' = \text{diag}(e^{2u_x}, e^{2u_y}, e^{2u_z})$, where $u_x, u_y, u_z$ are learnable, as in TLIO (Liu et al., 2020). Since our network predicts $\text{vec}(\Sigma) = \rho_\Sigma(F)\text{vec}(\Sigma')$, where both $\Sigma'$ and $F$ are learned, the resulting covariance is $\Sigma = F\Sigma' F^T$ (in matrix format). Via the transformation $F$ we can learn arbitrarily rotated $\Sigma$ in the $xy$-plane. We posit that this forces the frame network $\Psi$ to learn $F$ that aligns with the principle axes of the statistical uncertainty in displacement $\Sigma_d$. See App. A.2.3 for details on covariance parameterizations. Writing the singular value decomposition (SVD) we see that $\Sigma_d = U\text{diag}(\Sigma_{xx}, \Sigma_{yy}, \Sigma_{zz})U^T$. By inspection, this uncertainty is matched when $F$ aligns with principle directions $U$ and $\Sigma'$ aligns with the true uncertainties in those directions.

In the next section, let us now discuss the specific issue that arises when designing an equivariant frame network to process both $a_i$ and $\omega_i$, and the specific preprocessing step to remedy it.

## 4.2 DECOMPOSITION OF ANGULAR RATES

As previously discussed $a$ and $\omega$ transform under different representations $\rho_a \neq \rho_\omega$. This hinders joint feature learning since this would entail forming linear combinations of $a$ and $\omega$, and these linear combinations will not transform under $\rho_a$ or $\rho_\omega$. We propose a preprocessing step that decomposes $\omega_i$ into perpendicular vectors $v_{1,i}, v_{2,i}$ via a bijection $\mathcal{F} : \mathbb{R}^3 \to \mathbb{R}^3 \times \mathbb{R}^3$:

$$\mathcal{F}(\omega) = (v_1, v_2) = \left( \sqrt{\|\omega\|}\frac{w_1}{\|w_1\|}, \sqrt{\|\omega\|}\frac{w_2}{\|w_2\|} \right) \qquad \mathcal{F}^{-1}(v_1, v_2) = \omega = v_1 \times v_2 \tag{10}$$

We define $w_1 = [-\omega_y\, \omega_x\, 0]^T$ and $w_2 = \omega \times w_1$. If $\omega_x = \omega_y = 0$, we use $w_1 = a \times \omega$ and if both $\omega_x = \omega_y = 0$ and $a \times \omega = \mathbf{0}$, we use $w_1 = \omega \times [1\, 0\, 0]^T$.

Fig. 3 (b) shows that $v_1$ and $v_2$ transform with representation $\rho_{v_1}(F) = \rho_{v_2}(F) = F_{3\times3}$. Let variables with $*$ denote transformed vectors according to rotation $R$. Their cross product has the desirable property $\omega^* = v_1^* \times v_2^* = (R_{3\times3}v_1) \times (R_{3\times3}v_2) = \det(R_{3\times3})R_{3\times3}(v_1 \times v_2) = \det(R_{3\times3})R_{3\times3}\omega = \rho_\omega(R)\omega$, using the standard cross-product property $(Ax) \times (Ay) = \det(A)A(x \times y)$, and recalling that $R_{3\times3}$ is the matrix representation of $R$. The group action on $\omega$ exactly coincides with what was derived in Sec. 3. We only use this decomposition when with $O_g(3)$, where we process $a, v_1, v_2$ in a unified way. For $SO_g(3)$, we process $a, \omega$ which transform similarly since $\det(R_{3\times3}) = 1$.

## 4.3 TRANSITION TO $O(2)$ EQUIVARIANCE AND BASIC NETWORK LAYERS

Expressed in the gravity-aligned frame, representations $R_{3\times3}$ of $R \in O_g$ leave the z-axis unchanged, and can thus be decomposed into $R_{3\times3} = R_{2\times2} \oplus 1$, where the direct sum $\oplus$ constructs a block-diagonal matrix of its arguments. This decomposition motivates the decomposition $a_i = a_{i,xy} \oplus a_{i,z}$ and $v_{1/2,i} = v_{1/2,i,xy} \oplus v_{1/2,i,z}$, where $\oplus$ concatenates the $xy$-coordinates of each vector which transform with representation $R_{2\times2}$ and its $z$ component which transforms with representation 1, *i.e.* $z$ is invariant. This means that the $xy$-components transform according to representations of group $O(2)$ and implies that $O(2) \cong O_g(3)$. Inspired by Villar et al. (2021), we design our frame network to learn universally $O(2)$ equivariant outputs from invariant features alongside 2D vector features.

We convert the sequence of $n$ IMU measurements into $n \times C_0^s$ rotation invariant scalar features and $n \times 2 \times C_0^v$ vector features. As vector features we select the $xy$-components of each input

vector ($C_0^v = 2$ for $SO(2)$ corresponding with $a_{i,xy}, \omega_{i,xy}$ and $C_0^v = 3$ for $O(2)$ corresponding with $a_{i,xy}, v_{1,i,xy}, v_{2,i,xy}$). Instead, as scalar features we select *(i)* the $z$-components of each vector, *(ii)* the norm of the $xy$-components of each vector, and *(iii)* the pairwise dot-product of the $xy$-components of each vector. For $SO(2)$ we have $C_0^s = 2 + 2 + 1 = 5$, while for $O(2)$ we have $C_0^s = 3 + 3 + 3 = 9$. We process scalar features with MLPs and standard 1-D convolutions, vector features with specific linear and convolution layers, and combine both with specialized non-linear layers.

**Equivariant Linear Layer**  Following Villar et al. (2021), we design a $2D$ version of vector neuron (Deng et al., 2021) to process the vector features, enhancing efficiency. Following Finzi et al. (2021), we consider learnable linear mappings $v^{\text{out}} = W v^{\text{in}}$, with input and output vector features $v^{\text{in}}, v^{\text{out}} \in \mathbb{R}^2$ and seek a basis of weights $W \in \mathbb{R}^{2 \times 2}$, which satisfy $R_{2 \times 2} W v = W R_{2 \times 2} v$, i.e., equivariantly transform vector features $v \in \mathbb{R}^2$. This relation yields the constraint

$$(R_{2 \times 2} \otimes R_{2 \times 2}) \text{vec}(W) = \text{vec}(W), \tag{11}$$

Solving the above equation amounts to finding the eigenspace of the left-most matrix with eigenvalue 1. Such analysis for $R_{2 \times 2} \in SO(2)$ yields $W_{SO(2)} = w_1 I_{2 \times 2} + w_2 R_{90}$, where $R_{90}$ denotes a 90 degree counter-clockwise rotation in 2D, and $w_1, w_2 \in \mathbb{R}$ are learnable weights. Similarly, for $O(2)$ we find $W_{O(2)} = w_1 I_{2 \times 2}$. Vectorizing this linear mapping to multiple input and output vector features we have the following $SO(2)$ and $O(2)$ equivariant linear layers:

$$SO(2): \quad v^{\text{out}} = v^{\text{in}} W_1 + R_{90} v^{\text{in}} W_2 \qquad O(2): \quad v^{\text{out}} = v^{\text{in}} W_1 \tag{12}$$

with $v^{\text{in}} \in \mathbb{R}^{2 \times C_{\text{in}}}$, $v^{\text{out}} \in \mathbb{R}^{2 \times C_{\text{out}}}$ and $W_1, W_2 \in \mathbb{R}^{C_{\text{in}} \times C_{\text{out}}}$. Note that the $SO(2)$ layer has twice as many parameters as the $O(2)$ layer. We stack the above linear components into a kernel to design equivariant 1-D convolution layers. Since the IMU data forms a time sequence, we implement convolutions across time. We visualize our Linear and Convolutional Layers in Fig. 2.

**Nonlinear Layer**  Previous works (Weiler et al., 2018; Weiler & Cesa, 2019) propose various nonlinearities such as norm-nonlinearity, tensor-product nonlinearity, and gated nonlinearity for $SO(3)$ and $O(2)$ equivariance in an equivariant convolutional way; while Deng et al. (2021) applies per-point nonlinearity for vector features only. Since we already apply convolutions over time we simply apply a non-linearity pointwise. Unlike Deng et al. (2021), we need to mix scalar and vector features and thus adapt the gated nonlinearity (Weiler et al., 2018) to pointwise nonlinearity. Specifically, for $n$ vector and scalar features $v^{\text{in}} \in \mathbb{R}^{n \times 2 \times C}$, $s^{\text{in}} \in \mathbb{R}^{n \times C}$, we concatenate the norm features $\|v^{\text{in}}\| \in \mathbb{R}^{n \times C}$ with $s^{\text{in}}$. We run a single MLP with an output of size $n \times 2C$, and split it into new norm features $\gamma \in \mathbb{R}^{n \times C}$ and new activations $\beta \in \mathbb{R}^{n \times C}$ which we modulate with a non-linearity $s^{\text{out}} = \sigma(\beta)$. Finally, we rescale the original vector features according to the new norm:

$$\gamma, \beta = \text{mlp}(\|v^{\text{in}}\| \oplus_c s^{\text{in}}) \qquad v^{\text{out}} = \gamma v^{\text{in}} \qquad s^{\text{out}} = \sigma(\beta) \tag{13}$$

where $\oplus_c$ concatenates along the feature dimension. See Fig. 2b for more details.

## 5 EXPERIMENTS

We apply our framework to two types of neural inertial navigation systems: *(i)* an end-to-end deep learning approach (RONIN), and *(ii)* a filter-based approach with a learned prior (TLIO). Both networks process IMU samples in a gravity-aligned frame without gravity compensation, i.e., removing the gravity vector from the accelerometer reading. While RONIN regresses only a 2D velocity, TLIO estimates the orientation, position, velocity, and IMU biases using an EKF in 3D, which propagates states using raw IMU measurements and applies measurement updates with predicted displacement and uncertainty from a NN. Sec. 6 presents extensive ablations.

**Datasets:**  Our TLIO variant is trained on the TLIO Dataset (Liu et al., 2020) and tested on TLIO and Aria Everyday Activities (Aria) Datasets (Lv et al., 2024). Our RONIN variant is trained on RONIN Dataset (Herath et al., 2020). We train on the 50% open-sourced data. We test our RONIN variant on three popular pedestrian datasets RONIN (Herath et al., 2020), RIDI (Yan et al., 2018) and OxIOD (Chen et al., 2018b), which specifically target 2D trajectory tracking. RONIN-U (Unseen) contains IMU measurements from people who did not participate in the training and validation data collection. The people used to record RONIN-S (Seen) overlap with those from the training and validation set, but their data is disjoint from these sets. See App.A.3 for more dataset details, and App.A.5 and Fig. 2 for more details and visualizations of the equivariant network.

**Baselines:**  We compare EqNIO with TLIO with yaw augmentation (Liu et al., 2020), on the 3D benchmarks, and RONIN and RIO (Cao et al., 2022) on the 2D benchmarks. We also report the naive

| Model | TLIO Dataset | | | | | | Aria Dataset | | | | | |
|---|---|---|---|---|---|---|---|---|---|---|---|---|
| | MSE* $(10^{-2}m^2)$ | ATE $(m)$ | ATE* $(m)$ | RTE $(m)$ | RTE* $(m)$ | AYE (deg) | MSE* $(10^{-2}m^2)$ | ATE $(m)$ | ATE* $(m)$ | RTE $(m)$ | RTE* $(m)$ | AYE (deg) |
| TLIO | 3.333 | 1.722 | 3.079 | 0.521 | 0.542 | 2.366 | 15.248 | 1.969 | 4.560 | 0.834 | 0.977 | 2.309 |
| + rot. aug. | 3.242 | 1.812 | 3.722 | 0.500 | 0.551 | 2.376 | 5.322 | 1.285 | 2.103 | 0.464 | 0.521 | 2.073 |
| **+ SO(2) Eq. Frame** | 3.194 | 1.480 | 2.401 | 0.490 | 0.501 | 2.428 | 2.457 | 1.178 | 1.864 | 0.449 | 0.484 | 2.084 |
| **+ O(2) Eq. Frame** | 2.982 | 1.433 | 2.382 | 0.458 | 0.479 | 2.389 | 2.304 | 1.118 | 1.850 | 0.416 | 0.465 | 2.059 |

Table 1: Trajectory errors, lower being better. + rot. aug. is trained with yaw augmentations. Lowest, and second lowest values are marked in red and orange. * no EKF.

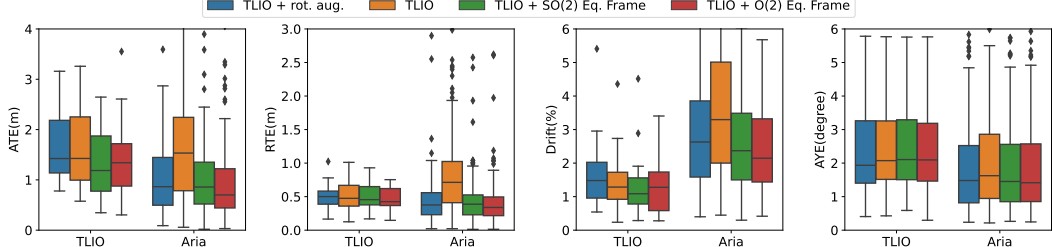

Figure 4: Trajectory errors for EqNIO applied to TLIO compared to vanilla TLIO trained with and without yaw augmentations on TLIO and Aria Datasets visualized with a box plot.

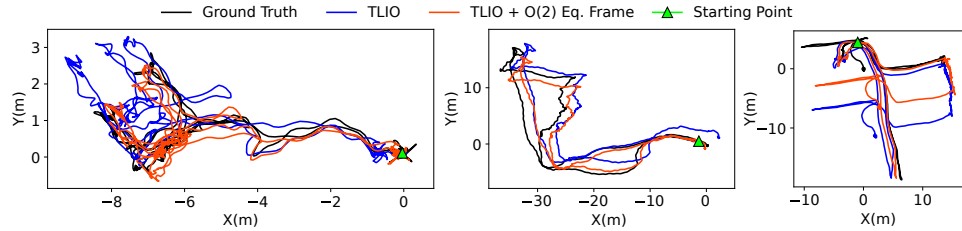

Figure 5: Groundtruth (black), and predicted trajectories on the TLIO Dataset by TLIO (Blue), and EqNIO (Red). Left and right have high, and the middle has medium difficulty.

IMU double integration (NDI) from Herath et al. (2020), and RONIN results trained on 100% of the RONIN training dataset. All other methods use only the public 50% of this dataset. RIO extends RONIN with two features: Joint optimization of an MSE loss on velocity predictions and cosine similarities with an equivariance constraint modeled using an auxiliary loss ($+J$) and an Adaptive Test-Time-Training strategy ($+TTT$). Finally, $+J+TTT$ combines both.

**Metrics:** The NN performance (indicated with $*$) is reported in terms of Mean Squared Error (MSE) in $10^{-2}m^2$, Absolute Translation Error (ATE) in $m$, and Relative Translation Error (RTE) in $m$, on the cumulative sum of the displacements predicted by the network as done in prior work. For TLIO, we also report the ATE in $m$, RTE in $m$, and Absolute Yaw Error (AYE) in degrees when running the EKF. See App. A.7 for more metric details. In what follows, we abbreviate EqNIO with +Eq. Frame.

## 5.1 RESULTS USING THE TLIO ARCHITECTURE

Tab. 1 compares baseline TLIO, trained with yaw augmentations as in (Liu et al., 2020) (+ rot.aug.), TLIO without augmentation (termed TLIO), and our two methods applied to TLIO without yaw augmentations (termed $+SO(2)$ and $+O(2)$ Eq. Frame). As seen in Tab. 1, $+O(2)$ Eq. Frame outperforms TLIO on metrics that ignore the EKF (with *) by a large margin of 57%, 12%, and 11% on MSE*, ATE*, and RTE* respectively. The $+SO(2)$ Eq. Frame model follows closely with 54%, 11%, and 7% respectively on the Aria Dataset. The performance of our methods is consistent across TLIO and Aria Datasets illustrating our generalization ability. Tab. 1 and Fig. 4 show our method surpasses the baseline on most metrics while remaining comparable in AYE. The superior performance of our model as compared to baseline TLIO when the NN is combined with EKF (*i.e.*, performance on ATE, RTE, and AYE metrics) is attributed to its generalization ability when the orientation estimate is not very accurate as well as the equivariant covariance predicted by the network. See Fig. 5 and App. A.8 for trajectory plots from the TLIO test dataset.

## 5.2 RESULTS USING THE RONIN ARCHITECTURE

On the RONIN dataset, we compare against RONIN and RIO† (indicated with +J, +TTT, and +J+TTT). RIO does not provide results for the RONIN Seen Dataset (RONIN-S) or RIDI Cross

| Model (RONIN) | RONIN-U ATE* (m) | RONIN-U RTE* (m) | RONIN-S ATE* (m) | RONIN-S RTE* (m) | RIDI-T ATE* (m) | RIDI-T RTE* (m) | RIDI-C ATE* (m) | RIDI-C RTE* (m) | OxIOD ATE* (m) | OxIOD RTE* (m) |
|---|---|---|---|---|---|---|---|---|---|---|
| + 100% data | 5.14 | 4.37 | 3.54 | 2.67 | 1.63 | 1.91 | 1.67 | 1.62 | 3.46 | 4.39 |
| + 50% data † | 5.57 | 4.38 | - | - | 1.19 | 1.75 | - | - | 3.52 | 4.42 |
| + 50% data + J † | 5.02 | 4.23 | - | - | 1.13 | 1.65 | - | - | 3.59 | 4.43 |
| + 50% data + TTT † | 5.05 | 4.14 | - | - | 1.04 | 1.53 | - | - | 2.92 | 3.67 |
| + 50% data + J +TTT † | 5.07 | 4.17 | - | - | 1.03 | 1.51 | - | - | 2.96 | 3.74 |
| + 50% data + **SO(2) Eq. Frame** | 5.18 | 4.35 | 3.67 | 2.72 | 0.86 | 1.59 | 0.63 | 1.39 | 1.22 | 2.39 |
| + 50% data + **O(2) Eq. Frame** | 4.42 | 3.95 | 3.32 | 2.66 | 0.82 | 1.52 | 0.70 | 1.41 | 1.28 | 2.10 |
| Naive Double Integration (NDI) | 458.06 | 117.06 | 675.21 | 1.6948 | 31.06 | 37.53 | 32.01 | 38.04 | 1941.41 | 848.55 |

Table 2: Trajectory errors with the RONIN architecture (lower is better), on the RONIN Unseen (-U), Seen (-S) (people used to record the test dataset are the same as train and validation), RIDI Test (-T), Cross Subject (-C), and the Oxford Inertial Odometry Datasets (OxIOD). Red (lowest), orange (second lowest). † results from Cao et al. (2022). * indicates no-EKF.

| Model | TLIO Dataset MSE* $(10^{-2}m^2)$ | ATE $(m)$ | ATE* $(m)$ | RTE $(m)$ | RTE* $(m)$ | AYE (deg) | Aria Dataset MSE* $(10^{-2}m^2)$ | ATE $(m)$ | ATE* $(m)$ | RTE $(m)$ | RTE* $(m)$ | AYE (deg) |
|---|---|---|---|---|---|---|---|---|---|---|---|---|
| TLIO | 3.333 | 1.722 | 3.079 | 0.521 | 0.542 | 2.366 | 15.248 | 1.969 | 4.560 | 0.834 | 0.977 | 2.309 |
| + rot. aug. | 3.242 | 1.812 | 3.722 | 0.500 | 0.551 | 2.376 | 5.322 | 1.285 | 2.102 | 0.464 | 0.521 | 2.073 |
| + rot. aug. + more layers | 3.047 | 1.613 | 2.766 | 0.524 | 0.519 | 2.397 | 2.403 | 1.189 | 2.541 | 0.472 | 0.540 | 2.081 |
| + rot. aug. + Non Eq. Frame | 3.008 | 1.429 | 2.443 | 0.495 | 0.496 | 2.411 | 2.437 | 1.213 | 2.071 | 0.458 | 0.508 | 2.096 |
| + rot. aug. + PCA Frame | 3.473 | 1.506 | 2.709 | 0.523 | 0.535 | 2.459 | 6.558 | 1.717 | 4.635 | 0.771 | 0.976 | 2.232 |
| + $SO(2)$ Eq. Frame + S | 3.331 | 1.626 | 2.796 | 0.524 | 0.536 | 2.440 | 2.591 | 1.146 | 2.067 | 0.466 | 0.517 | 2.089 |
| + $SO(2)$ Eq. Frame + P | 3.298 | 1.842 | 2.652 | 0.588 | 0.523 | 2.537 | 2.635 | 1.592 | 2.303 | 0.585 | 0.539 | 2.232 |
| + **SO(2) Eq. Frame** | 3.194 | 1.480 | 2.401 | 0.490 | 0.501 | 2.428 | 2.457 | 1.178 | 1.864 | 0.449 | 0.484 | 2.084 |
| + $O(2)$ Eq. Frame + S | 3.061 | 1.484 | 2.474 | 0.462 | 0.481 | 2.390 | 2.421 | 1.175 | 1.804 | 0.421 | 0.458 | 2.043 |
| + $O(2)$ Eq. Frame + P | 2.990 | 1.827 | 2.316 | 0.578 | 0.478 | 2.534 | 2.373 | 1.755 | 1.859 | 0.564 | 0.468 | 2.223 |
| + **O(2) Eq. Frame** | 2.982 | 1.433 | 2.382 | 0.458 | 0.479 | 2.389 | 2.304 | 1.118 | 1.849 | 0.416 | 0.465 | 2.059 |
| Eq CNN | 3.194 | 1.580 | 3.385 | 0.564 | 0.610 | 2.394 | 8.946 | 3.223 | 6.916 | 1.091 | 1.251 | 2.299 |

Table 3: Ablations with the TLIO architecture, lower is better. We test non-equivariant (+Non Eq. Frame), PCA-based (+PCA Frame), $SO(2)$ equivariant (+$SO(2)$ Eq. Frame), and $O(2)$ equivariant (+$O(2)$ Eq. Frame) frames, and yaw augmentation (+ rot. aug.). We also test $xy$-isotropic (+S) and Pearson-based (+P) covariance parameterizations. Eq CNN is a fully equivariant CNN. Red (lowest), orange (second lowest), and yellow (third lowest). * indicates no-EKF.

Subject Dataset (RIDI-C). As the RONIN base model does not use an EKF, we only report metrics with ∗. As Seen in Tab. 2, our methods significantly outperform the original RONIN by a large margin of 14% and 9% on ATE* and RTE* respectively even on the RONIN-U dataset. Our methods have better generalization as seen on RIDI-T and OxIOD, outperforming even +J+TTT† by a margin of 56% and 43% on ATE* and RTE* respectively on OxIOD Dataset. The +$O(2)$ Eq. Frame model converges at 38 epochs compared to over 100 in RONIN implying faster network convergence with our framework as compared to data augmentation. This demonstrates superior generalization of our strictly equivariant architecture. RIO's approach, involving multiple data rotations, test optimization, and deep ensemble at test time, would result in higher computational and memory costs as compared to our method. Finally, the comparison with NDI highlights the need for a neural displacement prior.

## 6 ABLATION STUDY

Here, we show the necessity of incorporating equivariance in inertial odometry, the choice of equivariant architecture, and covariance parameterization. We present all the ablations using the TLIO base model in Tab. 3, both with and without integrating the EKF. App. A.9 further contains the performance of all models above on a test dataset which is augmented with rotations and/or reflections. App. A.12, A.13, A.15, A.14 present sensitivity studies on the input sequence length, estimated gravity direction, IMU biases and IMU sampling rate.

***Baseline Ablation:*** **Is yaw augmentation needed when the input is in a local gravity-aligned frame?** We trained TLIO both with and without yaw augmentation using identical hyperparameters and the results in Tab. 3 (rows 1 and 2) reveal that augmentation enhances the network's generalization, improving all metrics for the Aria dataset with the lowest margin of 10% on AYE and highest margin of 65% for MSE*. This underscores the importance of equivariance for network generalization. **Does a Deeper TLIO with a comparable number of parameters match the performance of equivariant methods?** We enhanced the residual depth of the original TLIO architecture from 4

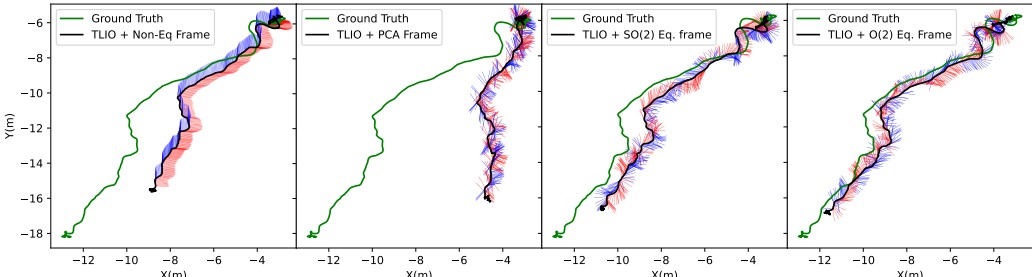

Figure 6: Trajectories and frame basis vectors (red/blue) on the Aria Dataset.

residual blocks of depth 2 each to depth 3 each (row 3) to match the number of parameters with our $+SO(2)$ Eq. Frame model (row 8). Despite having fewer parameters due to the removal of the orthogonal basis in $SO(2)$ vector neuron-based architecture, $+O(2)$ Eq. Frame model (row 11) still outperformed the deeper TLIO. The data from Tab. 3 show that merely increasing the network's size, without integrating true equivariance, is insufficient for achieving precise inertial odometry.

***Frame Ablation:*** **Can a non-equivariant MLP predict meaningful frames?** We trained TLIO with yaw augmentation and identical hyperparameters alongside an additional MLP mirroring the architecture of our method to predict a frame and term this baseline +Non Eq. Frame (row 4). We observed that +Non Eq. Frame tends to overfit to the TLIO dataset, and thus produce worse results on the Aria dataset. The predicted frames also poorly correlate with the underlying trajectory, as illustrated in Fig. 6. **Can frames predicted using PCA (handcrafted equivariant frame) achieve the same performance?** Using PCA to generate frames leads to underperformance on the Aria dataset, and worse results than the original TLIO which is likely due to PCA's noise sensitivity as shown in Fig. 6. Additionally, PCA cannot distinguish between $SO(2)$ and $O(2)$ transformations. Fig. 6 also shows that $O(2)$ does not have frames as smooth as $SO(2)$ as the reflected bends have reflected frames. See App. A.16 for an additional ablation using frame-averaging for canonicalization.

***Architecture Ablation:*** **Are fully equivariant architectures better ?** We trained a fully equivariant 1-D CNN using the layers in Sec. 4.3. Tab. 3 (row 12) shows our frame-based methods outperforming the equivariant CNNs, likely by leveraging the power of scalars and conventional backbones. We believe the fully equivariant architecture is overly restrictive, while also requiring a full network redesign. By contrast, our method can flexibly integrate existing state-of-the-art displacement priors.

***Covariance Ablation:*** **Do we need equivariant covariance?** We investigated the importance of equivariant covariance for both $SO(2)$ and $O(2)$ groups, as described in Sec. 4.1(See App. A.2.3 for covariance parameterizations). In Tab. 3, the models +S (rows 6 and 9) are trained with invariant covariance parameterized as $\Sigma = \text{diag}(e^{2u_x}, e^{2u_y}, e^{2u_z})$, that is unaffected by application of $F$. The results show that the equivariant covariance yields better performance, especially when combined with EKF, as it provides a more accurate estimate of the prediction covariance. **Can a full covariance matrix with Pearson parameterization improve performance?** In Tab. 3, our model outperforms +P (rows 7 and 10) in most cases indicating that by aligning the principal covariance axis into the basis of the equivariant frame, we intrinsically force covariance in the equivariant frame to be diagonal, which reduces ambiguity while training. Diagonal covariances improve convergence stability during optimization as stated in Liu et al. (2020). App. A.11 visualizes the covariance consistency of EqNIO and we conduct an analysis on the covariance parameterization in App. A.17.

## 7 CONCLUSION

We introduce a robust and generalizable neural displacement prior that combats drift in IMU-only neural inertial odometry via equivariant canonicalization. Our canonicalization scheme is generally applicable and eliminates the underlying yaw ambiguity in gravity-aligned frames which arise from roto-reflections in the plane around gravity. We fully characterize actions from this group on all relevant inputs and outputs of the prior and leverage this insight to design a network that produces learned frames that are $O_g(3)$ equivariant to these actions. By reducing the data variability seen by neural networks these frames boost the generalization of existing networks and enforce exact equivariance, unlike existing strategies that use data augmentation or equivariant consistency losses to enforce approximate equivariance. We demonstrate the generality of our framework through extensive validation on various datasets and applications to two base architectures (TLIO and RONIN). We believe this work paves the way for robust, and low-drift odometry running on edge devices.

ACKNOWLEDGEMENTS

We gratefully acknowledge support by the following grants: NSF FRR 2220868, NSF IIS-RI 2212433, ONR N00014-22-1-2677 and SNF 225354.

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

# A APPENDIX

## A.1 MORE RELATED WORKS

**Inertial Navigation Systems (INS):** The research in INS can be put into three broad categories: estimation-based techniques, multi-sensor data fusion (MSDF) techniques, and learning-based algorithms. The Kalman Filter (Kalman & Others, 1960) and it's variants, EKF, UKF and MEKF, have been implemented for inertial attitude estimation (Jing et al., 2017; Chiella et al., 2019; Hall et al., 2008), purely inertial odomety (Leishman et al., 2014; Titterton et al., 2004; Bortz, 1971), Visual-Inertial odometry (VIO) (Mourikis & Roumeliotis, 2007), LIDAR Odometry (Zhang & Singh, 2014; Zheng et al., 2024) and SLAM (Newman, 1999; Williams et al., 2000; Kim & Sukkarieh, 2003), but suffer from drift when using IMU data alone. Learning-based algorithms can reduce this drift, and have been designed for inertial attitude estimation (Weber et al., 2021; Brotchie et al., 2022; Asgharpoor Golroudbari & Sabour, 2023), inertial odometry (Chen et al., 2018a; Liu et al., 2020; Buchanan et al., 2023; Tang et al., 2024; Qiu et al., 2024), LIDAR odometry (Dong et al., 2024; Javanmard-Gh. et al., 2021; Okawara et al., 2024), VIO (Chen et al., 2021a; Pan et al., 2024) and SLAM (Wang et al., 2024; Peng et al., 2024). The consistency of VINS and EKF-SLAM has been studied in Hesch et al. (2014); Huang et al. (2008; 2009). Recent works have also explored equivariant observer design (Ng et al., 2023; Mahony et al., 2022; Bouazza et al., 2024), equivariant IMU Preintegration (Delama et al., 2024), the design of equivariant filters (van Goor et al., 2023; van Goor & Mahony, 2023; Fornasier et al., 2024) and geometric data fusion (Ge et al., 2024) for attitude estimation that do not include any learning-based components. They work on equivariant dynamics and use the lift function to map the state and the extended velocity to the Lie algebra associated with the system's symmetry group, enabling the construction of a lifted system on the Lie group. The error dynamics are linearized around a fixed origin, which makes it independent of the current state. To the best of our knowledge, we are the first to bake the equivariance into the NN for the inertial odometry problem. Integrating our equivariant network into these models would be potential future work.

**Canonicalization:** Both hand-crafted and learned canonicalization (Lowe, 2004; Yüceer & Oflazer, 1993; Jaderberg et al., 2015) has been widely studied and practiced. Recently, canonicalization has gained popularity in equivariant models, improving training efficiency and enabling seamless integration with standard model architectures. Kaba et al. (2023) first generally and theoretically proposed using Learned Canonicalization Functions to achieve equivariance. Puny et al. (2021) established standardizing the input by calculating the smallest set of frame set. Further, Ma et al. shows the connection between the frame-averaging method and canonical forms. In practical applications, Li et al. (2021); Baker et al. (2024); Luo et al. (2022); Du et al. (2022) canonicalize $3D$ point clouds by constructing global and local frames. Deng et al. (2021) introduces an equivariant canonicalization method in PointNet and Xu et al. (2024) integrates canonicalization into the Perceiver IO architecture. Additionally, with the rise of large pre-trained models, recent work (Mondal et al., 2023) applies canonicalization as a preprocessing step before inputting to such models.

## A.2 PRELIMINARY

### A.2.1 EQUIVARIANCE

In this section, we introduce more preliminaries of group and representation theory which form the mathematical tools for equivariance.

**Group** The group $G$ is a set equipped with an associative binary operation $\cdot$ which maps two arbitrary two elements in $G$ to an element in $G$. It includes an identity element, and every element in the set has an inverse element.

In this paper, we focus on the group $SO(2)$ and $O(2)$. $SO(2)$ is the set of all 2D planar rotations, represented by orthogonal matrices $R_{2\times 2} \in \mathbb{R}^{2\times 2}$ with $\det(R_{2\times 2}) = 1$. This group operation is matrix multiplication, and each rotation matrix has an inverse, which is its transpose. The identity element is the matrix representing no rotation.

$O(2)$ consists of all distance-preserving transformations in Euclidean 2D space, including both rotations and reflections. Elements of $O(2)$ are orthogonal matrices $R_{2\times2} \in \mathbb{R}^{2\times2}$ with $\det(R_{2\times2}) = \pm1$, with the group operation being matrix multiplication. Each transformation matrix has an inverse, and the identity element is the matrix representing no transformation.

**Group Representation and Irreducible Representation**   The group representation is a homomorphism from the group $G$ to the general linear map of a vector space $V$ of a field $K$, denoted $GL(V)$. In this work, we use $V = \mathbb{R}^n$ and $K = \mathbb{R}$

An irreducible representation (irrep) of a group $G$ is a representation in which the only invariant subspaces under the action of $G$ are the trivial subspace $\{\mathbf{0}\}$ and the entire space $V$. In other words, an irreducible representation cannot be broken down into smaller, nontrivial representations,i.e., it cannot be the direct sum of several nontrivial representations.

For $SO(2)$, we can use $\theta \in (0, 2\pi]$ to represent $SO(2)$, for any $\theta$, the irreducible representation of the frequency $n \in \mathbb{N}$ is:

$$\rho_n(\theta) = \begin{pmatrix} \cos n\theta & -\sin n\theta \\ \sin n\theta & \cos n\theta \end{pmatrix}.$$

In this work, we use $n = 1$. For $O(2)$, we can use $r \in \{-1, 1\}$ to denote reflection and $\theta \in (0, 2\pi]$ to denote rotation. The trivial representation $\rho_0(r, \theta) = 1$. For the nontrivial representation of frequency $n \in \mathbb{N}^+$

$$\rho_n(r, \theta) = \begin{pmatrix} \cos(n\theta) & -\sin(n\theta) \\ \sin(n\theta) & \cos(n\theta) \end{pmatrix} \begin{pmatrix} 1 & 0 \\ 0 & r \end{pmatrix}$$

There is another one-dimensional irrep for $O(2)$, $\rho(r, \theta) = r$ which corresponds to the trivial representation of rotation.

The introduction to group representations has been covered extensively in previous work on equivariance (Cohen & Welling, 2016; Weiler et al., 2018; Xu et al., 2024). Specifically, for $SO(2)$ and $O(2)$, Weiler & Cesa (2019) provide a detailed introduction.

**Invariance and Equivariance**   Given a network $\Phi : \mathcal{X} \to \mathcal{Y}$, if for any $x \in \mathcal{X}$,

$$\Phi(\rho^{\mathcal{X}} x) = \Phi(x),$$

implies the group representation $\rho^{\mathcal{Y}}$ of the output space is trivial, *i.e.* identity, and the input does not transform (*i.e.* the input is invariant) under the action of the group. Note that in our paper $\mathcal{X}$ and $\mathcal{Y}$ can be seen as the direct sum (concatenation) of all input and output vectors respectively. In our paper, the coordinates/ projections of $3D$ vector to the gravity axis $z$-axis are invariant, therefore we call them invariant scalars.

A network $\Phi : \mathcal{X} \to \mathcal{Y}$ is equivariant if it satisfies the constraint

$$\Phi(\rho^{\mathcal{X}} x) = \rho^{\mathcal{Y}} \Phi(x).$$

In this paper, the input is the sequence of accelerations and angular velocities, and the output is composed of displacement and covariance. For displacement, the $z$-component is invariant while $xy$-components are acted under the representation of $\rho_1$ defined in the above section. Hence, for displacement, $\rho^{\mathcal{Y}} = \rho_1 \oplus 1$ and for covariance $3D$ covariance, $\rho^{\mathcal{Y}} = (\rho_1 \oplus 1) \otimes (\rho_1 \oplus 1)$

**Subequivariance**   As mentioned in prior works (Chen et al., 2023; Han et al., 2022), the existence of gravity breaks the symmetry in the vertical direction, reducing O(3) to its subgroup O(2). We formally characterize this phenomenon of equivariance relaxation as subequivariance. We have mathematically defined the subequivariance in Section 3 of the paper. In simpler terms, the gravity axis is decoupled and treated as an invariant scalar while the other two axes are handled as a separate 2D vector. Upon rotation, the invariant scalar remains constant while the other two axes are transformed under rotation. So we are limited now to SO(2) rotations and roto-reflections. In the general case of equivariance, the 3D vector would be considered three-dimensional and an SO(3) rotation would act on it. The transformation would be along all three axes.

A.2.2   INERTIAL ODOMETRY

In this section, we introduce more preliminaries on the terms used in inertial odometry.

**Inertial Measurement Unit**   Inertial Measurement Unit (IMU) is an electronic device that measures and reports linear acceleration, angular velocity, orientation, and other gravitational forces. An IMU typically consists of a 3-axis accelerometer, a 3-axis gyroscope, and depending on the heading requirement a 3-axis magnetometer.

An accelerometer measures instantaneous linear acceleration ($a_i$). It can be thought of as a mass on a spring, however in micro-electro-mechanical systems (MEMS) it is beams that flex instead of spring.

A gyroscope measures instantaneous angular velocity ($\omega_i$). It measures the angular velocity of its frame, not any external forces. Traditionally, this can be measured by the fictitious forces that act on a moving object brought about by the Coriolis effect, when the frame of reference is rotating. In MEMs, however, we use high-frequency oscillations of a mass to capture angular velocity readings by the capacitance sense cones that pick up the torque that gets generated.

**World Frame**   A world frame, also known as a cartesian coordinate frame, is a fixed frame with a known location and does not change over time. In this paper, we denote the fixed frame with $z$-axis perfectly aligned with the gravity vector as the world frame, denoted as $w$.

**Local-gravity-aligned Frame**   A local-gravity-aligned frame has one of its axes aligned with the gravity vector at all times but it is not fixed to a known location.

**Body Frame**   A body frame comprises the origin and orientation of the object described by the navigation solution. In this paper, the body frame is the IMU's frame. This is denoted as $b$ for the IMU data.

**Gravity-compensation**   Gravity compensation refers to the removal of the gravity vector from the accelerometer reading.

**Gravity-alignment**   Gravity-alignment of IMU data refers to expressing the data in the gravity-aligned frame. This is done by aligning the $z$-axis of the IMU inertial frame with the gravity vector pointing downwards and is usually achieved by fixing the roll and pitch (rotations around the $x$ and $y$ axes) or by applying a transformation estimated by the relative orientation between the gravity vector and a fixed $z$-axis pointing downwards. This is usually achieved with a simple rotation.

A.2.3   UNCERTAINTY QUANTIFICATION IN INERTIAL ODOMETRY

In this section, we provide more context on uncertainty quantification in odometry and detail the different parameterizations used for regressing the covariance matrix in the paper.

**Homoscedastic Uncertainty**   Homoscedastic uncertainty refers to uncertainty that does not vary for different samples, i.e., it is constant.

**Heteroscedastic Uncertainty**   Heteroscedastic uncertainty is uncertainty that is dependent on the sample, i.e., it varies from sample to sample.

**Epistemic Uncertainty**   Epistemic uncertainty is uncertainty in model parameters. This can be reduced by training the model for longer and/or increasing the training dataset to include more diverse samples.

**Aleatoric Uncertainty**   Aleatoric uncertainty is the inherent noise of the samples. This cannot be reduced by tuning the network or increasing the diversity of the data.

**Why do we need to estimate uncertainty in inertial odometry?**   Inertial odometry Kalman filters need measurement and plant covariances that are traditionally manually tuned and are considered

constant cross-measurement (homoscedastic). Neural networks offer us a way to provide an uncertainty estimate for each measurement. In the case of TLIO, the measurement of the Kalman Filter is a displacement and the neural network predicts both the measurement and its covariance.

**What is the uncertainty we are estimating in inertial odometry?**    We are regressing the aleatoric uncertainty using the neural network.

**How is the uncertainty estimated in this paper?**    We regress aleatoric uncertainty as a covariance matrix jointly while regressing 3D displacement following the architecture of TLIO (Liu et al., 2020). Since there is no ground truth for the covariance, we use the negative log-likelihood loss of the prediction using the regressed Gaussian distribution. As this loss captures the Mahalanobis distance, the network gets jointly trained to tune the covariance prediction.

**Diagonal covariance matrix**    TLIO (Liu et al., 2020) regresses only the three diagonal elements of the covariance matrix as $\log \sqrt{\Sigma_{xx}}$, $\log \sqrt{\Sigma_{yy}}$ and $\log \sqrt{\Sigma_{zz}}$ and the off-diagonal elements are zero. This formulation constrains the uncertainty ellipsoid to be along the local gravity-aligned frame.

**Full covariance matrix using Pearson correlation**    Russell & Reale (2021) define a parameterization to regress the full covariance matrix. They regress six values of which three are the diagonal elements $\log \sqrt{\Sigma_{xx}}$, $\log \sqrt{\Sigma_{yy}}$ and $\log \sqrt{\Sigma_{zz}}$ and the remaining three are Pearson correlation coefficients $\rho_{xy}$, $\rho_{yz}$, and $\rho_{xz}$. The diagonal elements are obtained by exponential activation while the off-diagonal elements are computed as follows

$$\Sigma_{ij} = \rho_{ij} \sqrt{\Sigma_{ii} \Sigma_{jj}}$$

where $\rho_{ij}$ passes through tanh activation.

**Diagonal covariance matrix in canonical frame**    In our approach, we regress the three diagonal elements as $\log \sqrt{\Sigma_{xx}}$, $\log \sqrt{\Sigma_{yy}}$ and $\log \sqrt{\Sigma_{zz}}$ in the invariant canonical frame. Since the $z$-axis is decoupled from the $xy$-axis, only $\Sigma_{xx}$ and $\Sigma_{yy}$ are mapped back using the equivariant frame to obtain a full 2D covariance matrix from the diagonal entries. The resulting matrix is as follows

$$\begin{bmatrix} \Sigma_{xx} & \Sigma_{xy} & 0 \\ \Sigma_{xy} & \Sigma_{yy} & 0 \\ 0 & 0 & \Sigma_{zz} \end{bmatrix}$$

## A.3    DATASET DETAILS

In this section, we provide a detailed description of the 4 datasets used in this work - TLIO and Aria for TLIO architecture, and RONIN, RIDI and OxIOD for RONIN architecture.

**TLIO Dataset-**    The TLIO Dataset (Liu et al., 2020) is a headset dataset that consists of IMU raw data at 1kHz and ground truth obtained from MSCKF at 200 Hz for 400 sequences totaling 60 hours. The ground truth consists of position, orientation, velocity, IMU biases and noises in $\mathbb{R}^3$. The dataset was collected using a custom rig where an IMU (Bosch BMI055) is mounted on a headset rigidly attached to the cameras. This dataset captures a variety of activities including walking, organizing the kitchen, going up and down stairs, on multiple different physical devices and more than 5 people for a wide range of individual motion patterns, and IMU systematic errors. We use their data splits for training (80%), validation (10%), and testing(10%).

**Aria Everyday Dataset-**    Aria Everyday Dataset (Lv et al., 2024) is an open-sourced egocentric dataset that is collected using Project Aria Glasses. This dataset consists of 143 recordings accumulating to 7.3 hrs capturing diversity in wearers and everyday activities like reading, morning exercise, and relaxing. There are two IMUs on the left and right side of the headset of frequencies 800 and 1kHz respectively. They have two sources of ground truth- open and closed loop trajectory at 1kHz. Open loop trajectory is strictly causal while closed loop jointly processes multiple recordings to place them in a common coordinate system. The ground truth contains position and orientation in $\mathbb{R}^3$. We use it as a test dataset. The raw right IMU data is used to compare closed-loop trajectory with EKF results. The data was downsampled to 200Hz and preprocessed using the closed-loop trajectory to test the NN trained on TLIO.

**RONIN Dataset-** RONIN Dataset (Herath et al., 2020) consists of pedestrian data with IMU frequency and ground truth at 200Hz. RONIN data features diverse sensor placements, like the device placed in a bag, held in hand, and placed deep inside the pocket, and multiple Android devices from three vendors Asus Zenfone AR, Samsung Galaxy S9, and Google Pixel 2 XL. Hence, this dataset has different IMUs depending on the vendor. We use RONIN data splits to train and test their model with and without our framework. RONIN-U (Unseen) is a test set that contains IMU measurements from people who did not participate in the training and validation data collection. The people used to record the RONIN-S (Seen) overlap with those from the training and validation set, but their data is disjoint from the training and validation set. The RONIN-U dataset, thus, tests the generalization capabilities of a method.

**RIDI Dataset-** RIDI Dataset (Yan et al., 2018) is another pedestrian dataset with IMU frequency and ground truth at 200 Hz. This dataset features specific human motion patterns like walking forward/backward, walking sidewards, and acceleration/deceleration. They also record data with four different sensor placements. We report test results of RONIN models on both the RIDI test and cross-subject datasets. RIDI results are presented after post-processing the predicted trajectory with the Umeyama algorithm (Umeyama, 1991) for fair comparison against other methods.

**OxIOD Dataset-** OxIOD Dataset (Chen et al., 2018b) stands for Oxford Inertial Odometry Dataset consists of various device placements/attachments, motion modes, devices, and users capturing everyday usage of mobile devices. The dataset contains 158 sequences totaling 42.5 km and 14.72 hours captured in a motion capture system. We use their unseen multi-attachments test dataset for evaluating our framework applied to RONIN architecture.

## A.4    COMPARISON WITH PREVIOUS WORK

We start off by describing the different types of equivariance used in neural inertial odometry, followed by an in-detail description of each of the related works our method is compared.

### A.4.1    COMPARISON OF TYPES OF EQUIVARIANCE

In the current literature on learning-based purely inertial odometry, there are primarily three ways of achieving equivariance: (1) data augmentation, (2) the imposition of equivariant constraints in the form of auxiliary losses, and (3) the tailoring of NN to respect exact equivariance *by design*. The latter is the strategy used in this work.   These different types of equivariance used in inertial

| | Transformation Group | Training Loss | Training Input Augmentation | Training GT Augmentation | Test Input Augmentation | Strict/Approximate Equivariance |
|---|---|---|---|---|---|---|
| TLIO Liu et al. (2020) | $SO(2)$ | MLE | Yes | Yes | No | Approximate |
| RONIN Herath et al. (2020) | $SO(2)$ | MSE | Yes | Yes | No | Approximate |
| RIO Cao et al. (2022) | $C_4$ | MSE + Auxiliary Equivariance Loss | Yes | No | Yes | Approximate |
| Ours | $O(2)$ | MLE | No | No | No | Stricts |

Table 4: Comparison of our method with previous methods from the perspective of equivariance

odometry are illustrated in Figure 7.  Importantly, while data-augmentation and auxiliary losses enforce approximate equivariance, leading to residual equivariance errors even after training, the use of equivariant NNs enforces strict equivariance, and thus equivariance error.

### A.4.2    TLIO

TLIO, described in  (Liu et al., 2020), consists of two components:  an Extended Kalman Filter (EKF), to estimate the current orientation, position, velocity, and IMU biases, and a NN to predict a displacement prior, which is defined via its mean displacement and associated uncertainty.

We treat the displacement prior as a virtual measurement (with associated realization and measurement noise given by the mean displacement and covariance) and fuse it with the EKF state at 20 Hz. The EKF details can be found in Appendix A.6. At the IMU sampling rate (1kHz for the TLIO and

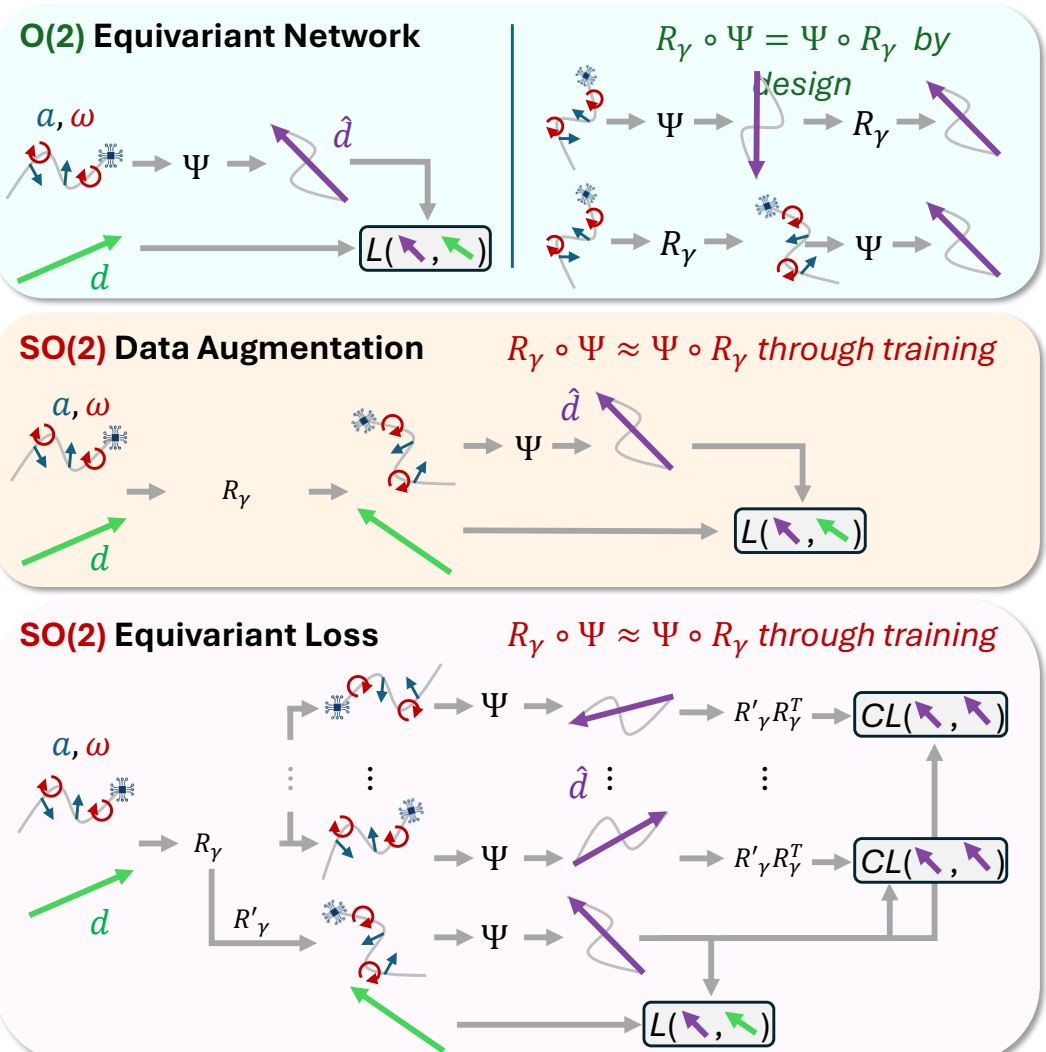

Figure 7: Overview of different equivariance types in neural inertial odometry. Works can be split into those using $SO(2)$ Data augmentation, such as Liu et al. (2020); Herath et al. (2020); Cao et al. (2022), those using $SO(2)$ Equivariant losses Cao et al. (2022), and our work, which tackles this challenge with an $O(2)$ equivariant network. Data augmentation methods apply random yaw ($R_\gamma$) on acceleration $a$, angular velocity $\omega$, and displacements $d$ for each loaded sample, and thus achieve approximate equivariance to yaw rotations ($\rho(R_\gamma) \circ \Psi \approx \Psi \circ \rho(R_\gamma)$). By contrast, methods that use $SO(2)$ equivariant losses pass multiple randomly rotated samples through the network and enforce consistency between respective outputs using a self-supervised equivariant loss ($CL$). Finally, our method employs an equivariant canonicalization scheme which employs specialized equivariant layers that render our network inherently exactly equivariant ($\rho(R_\gamma) \circ \Psi = \Psi \circ \rho(R_\gamma)$) to roto-reflection elements $R_\gamma \in O(2)$ (a more general group than $SO(2)$). Our network thus does not require data augmentation, or expensive equivariant losses and still achieves better results.

Aria Dataset), the EKF state and covariance are propagated using the IMU measurements. Since the measurement equation depends on current and past states, we initiate EKF state augmentation and cloning at each measurement. To avoid yaw observability issues, we use a strategy similar to Chen et al. (2022) to linearize at the beginning of the 1-second window of past update/clone states. The EKF current states at the update frequency are stored as clone states before the EKF measurement update step. This is referred to as state augmentation, and the augmented past states are referred to as clone states.

The neural displacement prior $\hat{d}$ and the associated prediction covariance $\hat{\Sigma}$ are predicted from a NN. The NN takes as input IMU samples linearly interpolated from 1kHz to 200Hz. This implies that within a time window of 1 second, the network takes as input 200 IMU samples of 6 dimensions of which 3 dimensions correspond to accelerometer readings and 3 correspond to gyroscope readings. The input samples are first expressed in the gravity-aligned world frame by estimating the IMU orientation using the noisy gyroscope measurements. Finally, they are expressed in the local gravity-aligned frame using the yaw-estimate from the EKF at the clone state $i$ corresponding to the beginning of the 1-second window. The NN input is bias compensated using factory calibration values of IMU accelerometer and gyroscope biases and does not use the EKF bias estimates, to avoid instabilities. The predicted covariance is parametrized as a diagonal covariance matrix and claimed in TLIO to lead to better convergence and stability during training. The NN is trained with VIO estimates of bias and orientation and uses displacement calculated from VIO position estimates as supervision data.

During deployment with the EKF, *(i)* bias estimates may not be accurate as they are set to factory calibration values, and *(ii)* gravity orientation may be noisy due to noise in orientation estimation. These noise sources lead to drift. To mitigate this drift, during training TLIO uses three data augmentation strategies: *(i)* IMU bias augmentation using $n_a \sim \mathcal{U}[0.2, 0.2]m/s^2$ additive noise for the accelerometer bias and $n_\omega \sim \mathcal{U}[0.05, 0.05]rad/s$ additive noise for the gyroscope bias. *(ii)* Gravity direction perturbation rotates the samples around a random horizontal rotation axis by an angle $\alpha \sim \mathcal{U}[0, 5]deg$. *(iii)* Yaw ($SO(2)$) augmentation rotates the network inputs and ground truth by a random yaw angle $\gamma \sim \mathcal{U}[-\pi, \pi]$.

The network is trained with a batch size of 1024 samples for 50 epochs, with 1770 iterations per epoch. The NN architecture as presented in Liu et al. (2020) consists of a 1D convolutional ResNet consisting of dilated causal convolutions followed by two disjointed 3-layer MLPs after flattening the ResNet output. One head corresponds to the 3D displacement prediction and the other head corresponds to 3 diagonal covariance values. The NN is trained for 10 epochs with a Mean Squared Error loss and the remaining 40 epochs with a Mean Log Likelihood Error loss (MLE) as we want the displacement prediction to converge before indirect supervision of the predicted covariance.

### A.4.3 RONIN

RONIN is the predecessor of TLIO and is an end-to-end architecture regressing 2D velocity from IMU samples over 1-second windows. The RONIN training dataset consists of pedestrian data which has 2D position as ground-truth (GT). The GT displacement is divided by the time interval to obtain ground truth velocity for training. The regressed 2D velocities are directly integrated to obtain 2D position estimates.

During evaluation the predicted and ground truth trajectories are aligned using the Umeyama Umeyama (1991) (scaled Procrustes) algorithm, since RONIN does not estimate orientation. The NN architecture of RONIN is the same as TLIO where the network takes 200 IMU inputs of dimension 6 and consists of 1D convolutional ResNet architecture followed by a single 3-layer MLP head to regress 2 values corresponding to the x-axis and y-axis of the predicted velocity. As there is no EKF, they do not estimate covariance. The only augmentation done is yaw augmentation similar to TLIO. The random yaw angles sampled from $\mathcal{U}[0, 2\pi]$ are used to rotate a single batch of input samples and corresponding GT velocities. The batch size is 1024, and the network is trained for 100 epochs. The NN is trained only on the MSE loss.

### A.4.4 RIO

RIO proposes a framework that can be easily integrated with existing learning-based inertial odometry architectures. During training, RIO computes an auxiliary equivariant loss in the following way. It passes the network input and a randomly yaw-rotated copy (yaw rotation from $SO(2)$ sampled from $\mathcal{U}[0, 2\pi]$) through the neural network, and computes two losses. The first compares the output from the original sample to ground truth (via an MSE loss), and the second measures the negative cosine similarity between the original prediction and the prediction obtained from rotated input. This training procedure is termed the joint-training strategy. When the magnitude of ground truth velocity is less than 0.5 m/s (termed as nearly stationary), the auxiliary loss is ignored. During inference, RIO uses a prediction model, and a model ensemble to employ an adaptive test-time-training strategy. The model ensemble is used to estimate the uncertainty in prediction and trigger retraining of the prediction model. This ensemble consists of NNs that were trained from different random initializations. The variance of the predictions from the models is considered as the uncertainty of the predictions. Retraining occurs with a history of 128 samples, by minimizing the auxiliary loss and using the prediction model for initialization. Each of the samples is rotated by the four angles $\beta \in [72, 144, 216, 288]$ degrees. When the model predicts velocities below 0.5 m/s the prediction model is reset. So at any instance of time, there is one copy of the model that is continuously tuned during inference, one copy of the original trained network, and a set of randomly initialized networks to get an ensemble-based uncertainty estimate.

While our work and RIO both aim to address the equivariance of the neural inertial odometry by modeling rigid transformations of trajectory and IMU data, there are several distinct differences. Most significantly, RIO addresses *approximate* equivariance, while our method enforces *strict equivariance*. Approximate equivariance enforces equivariance via an equivariance loss which penalizes inconsistencies in predictions on data from four rotated trajectories. By contrast, *strict* equivariance enforces *inherent* consistency by tailoring specific NN components to be *exactly* equivariant. This work proposes several specialized linear and non-linear layers to guarantee the strict equivariance of the predicted canonical frame, while RIO uses traditional layers.

RIO adopts a test-time training strategy, which adds an additional computational overhead and thus increases the inference time as they are required to store multiple versions of the prediction model and augmented trajectories. Since our method is strictly equivariant, we do not require a Test-Time Training strategy. Moreover, despite not using such a strategy our method outperforms RIO.

Finally, RIO only handles $2D$ displacement outputs and equivariance to rotations in SO(2), i.e. $2D$ planar rotations. By contrast, our method can handle both $2D$ and $3D$ outputs and thus addresses equivariance in $2D$ and subequivariance in $3D$. Moreover, we also extend the modeling to roto-reflections, i.e., the group $O(2)$, which comprises rotations and reflections in the plane perpendicular to gravity and requires a novel bijection for angular rate preprocessing.

## A.5 EQUIVARIANT NETWORK IMPLEMENTATION DETAILS

In this section, we first provide an overview of the canonical displacement prior and then describe in detail the equivariant network implementation and how it is combined with TLIO and RONIN.

### A.5.1 OVERVIEW OF CANONICAL DISPLACEMENT PRIOR

(1) Following Figure 2 we first gravity align the IMU data, i.e. rotate it such that the z-axis is aligned with gravity (estimated online by an EKF).

(2) We then process this IMU data by an "equivariant frame model" that outputs two vectors. This network takes in vector and scalar features derived from the IMU data. The equivariant network is strictly O(2) or SO(2) equivariant by design. The predicted vectors are then converted into an orthogonal set of unit vectors using Gram-Schmidt orthogonalization.

(3) Then, we transform the input IMU data using the equivariant frame from (2) ("canon." block in Figure 2) to produce invariant (consistent) inputs within the canonical frame, ensuring robustness to any roto-reflection group transformations applied to the original data.

(4) The invariant input is fed into a standard neural network architecture ("off-the-shelf model"), such as TLIO's ResNet, to generate a displacement and covariance, termed canonical displacement prior expressed with respect to the learned canonical frame. These outputs remain consistent under transformations from the roto-reflection group applied to the original input.

(5) Lastly, we project back the predicted displacement and covariance using the canonical frame to obtain an equivariant displacement and covariances using the equivariant frame of (2).

Finally, depending on the backbone architecture on which our framework is applied (in our case for TLIO), the predicted displacement and covariances are fed into a filtering algorithm, like an EKF as a measurement to update the estimate of the IMU state (orientation, position, velocity and biases).

### A.5.2    IMPLEMENTATION DETAILS

The input to the framework is IMU samples from the accelerometer and gyroscope for a window of 1 second with IMU frequency 200Hz resulting in $n = 200$ samples. All IMU samples within a window are gravity-aligned with the first sample at the beginning of the window, previously referred to as the clone state. The network design, as seen in Figure 2 b, differs in architecture for $SO(2)$ and $O(2)$ and hence is described separately below.

$SO(2)$**-**   We decouple the $z$-axis from the other two axes and treat linear acceleration and angular velocity along the $z$-axis as scalars (2). We also take the norm of the 2D accelerometer and gyroscope measurements (2), their inner product (1) resulting in invariant scalars $\mathbb{R}^{n \times 5}$. The $x$ and $y$ components of IMU measurements are passed as vector inputs $\mathbb{R}^{n \times 2 \times 2}$. The vectors and scalars are then separately passed to the linear layer described in Section 4.3. The equivariant network predicting the equivariant frame consists of 1 linear layer, 1 nonlinearity, 1 convolutional block with convolution applied over time, non-linearity, and layer norm. The hidden dimension is 128 and the convolutional kernel is 16 x 1. Finally, the fully connected block of hidden dimension 128 consisting of linear, nonlinearity, layer norm, and output linear layer follows a pooling over the time dimension. The output of the final linear layer is 2 vectors representing the two bases of the equivariant frame. The input vectors of dimension $\mathbb{R}^{n \times 2 \times 2}$ are projected into the invariant space via the equivariant frame resulting in invariant features in $\mathbb{R}^{n \times 4}$. These features are combined with the input scalars and passed as input ($\mathbb{R}^{n \times 6}$) to TLIO or RONIN base architecture. The output of TLIO is invariant 3D displacement and diagonal covariance along the principal axis. The output of RONIN is 2D velocity. The $x$ and $y$ components are back-projected using the equivariant frame to obtain displacement vector $d$ in $\mathbb{R}^2$ and the covariance in the original frame. The covariance is parameterized and processed as mentioned in Section 4.1.

$O(2)$**-**   The preprocessing is as described in Section 4.1 where $\omega$ is decomposed to two vectors $v_1$ and $v_2$ that each have magnitude $\sqrt{\|\omega\|}$. The preprocessed input therefore consists of 3 vectors $a$, $v_1$, and $v_2$. This is then passed to the equivariant network by decoupling the $z$-axis resulting in vector input $\mathbb{R}^{n \times 3 \times 2}$ which represents 3 vectors in 2D. The scalars passed to the linear layer described in Section 4.3 consist of the accelerometer $z$-axis measurement (1), the $z$ component of the two vectors $v_1$ and $v_2$ (2), the norm of the vectors (3) and the inner product of the vectors(3) resulting in $\mathbb{R}^{n \times 9}$. The network architecture is the same as $SO(2)$ with hidden dimension 64 and 2 convolutional blocks in order to make it comparable in the number of parameters to $SO(2)$ architecture. The invariant features obtained by projecting the three vectors using the equivariant frame are processed as mentioned in Section 4.1 to obtain 2 vectors in 3D that are fed as input to TLIO and RONIN. The postprocessing is the same as $SO(2)$.

The framework is implemented in Pytorch and all hyperparameters of the base architectures are used to train TLIO and RONIN respectively. The $SO(2)$ architecture has 8,884,870 while $O(2)$ has 6,020,230 parameters and the base TLIO architecture has 5,424,646. The baseline TLIO and our methods applied to TLIO were trained on NVIDIA a40 GPU occupying 7-8 GB memory per epoch. The training took 5 mins per epoch over the whole training dataset. We train for 10 epochs with MSE Loss and the remaining 40 epochs with MLE Loss similar to TLIO (Liu et al., 2020). RONIN was trained on NVIDIA 2080ti for 38 epochs taking 2 mins per epoch. The loss function used was MSE as mentioned in Herath et al. (2020). The EKF described in TLIO was run on NVIDIA 2080ti with the same initialization and scaling of predicted measurement covariance as in TLIO (Liu et al., 2020).

We compare the resource requirements of the SO(2), and O(2) variant of our method coupled with TLIO, with base TLIO without an equivariant frame. We report the floating point operations (FLOPs), the inference time (in milliseconds), and Maximum GPU memory (in GB) during inference, on an NVIDIA 2080 Ti GPU for the NN averaged over multiple runs to get accurate results. While base TLIO uses 35.5 MFLOPs, 2.79 ms, and 0.383 GB per inference, our SO(2) equivariant method instead uses 531.9 MFLOPs, 5.43 ms, and 0.383 GB per inference. Finally, our O(2) equivariant method uses 638.5 MFLOPs, 5.70 ms, and 0.385 GB per inference. We further evaluate the Maximum GPU memory for the equivariant networks separately and report 0.255 GB per inference for SO(2) equivariant frame prediction and 0.257 GB per inference for O(2) equivariant frame prediction. The Maximum GPU memory is unaffected because the equivariant frame computation utilizes less memory than TLIO.

Finally, we also evaluate our method with a downstream EKF on an NVIDIA 2080 Ti GPU. The EKF incorporates raw IMU measurements for propagation, and displacement measurements from the NN as measurement updates. For every 20 imu samples, we send the last 200 IMU measurements to the NN to provide this measurement update. The original TLIO requires 0.492 seconds and 1.113 GB of memory. For the SO(2) variant of our method, we require 0.554 seconds and 1.109 GB of memory to process 1 second of real-world data. For the O(2) variant, we use 0.554 seconds and 1.115 GB of memory, showing that our method is faster than real-time. The increase in memory for the O(2) variant is due to the additional preprocessing step.

With comparable computing resources, our equivariant model outperforms TLIO since we leverage symmetry, which is an intrinsic property in inertial odometry.

### A.6 EKF DETAILS

The EKF continuously estimates the orientation, linear velocity, position, acceleration, and gyroscope biases. In TLIO architecture, the EKF propagates the IMU samples at a higher frequency of 1 kHz while the NN takes clone states at 200Hz to predict the displacement over 1 second time window and its associated uncertainty. Throughout the paper, we denote the clone state indices with i and the EKF propagation indices with k. We further outline the EKF state definition, propagation model, state augmentation, measurement, and update model below for completeness of the manuscript. It must be noted that we follow TLIO (Liu et al., 2020) and do not make any modifications to this part of the architecture.

#### A.6.1 EKF STATE DEFINITION

The state of the EKF is defined as $X = (\xi_1, ...., \xi_n, s)$ where $\xi_i, i = 1, ..., n$ represents the $n$ clone states whose corresponding IMU measurements are passed to the NN as input and $s$ has the current propagation state.

$$\xi_i = ({}^w_b R_i, {}^w p_i) \quad s = ({}^w_b R, {}^w v, {}^w p, b_g, b_a)$$

where ${}^w_b R_i$ represents the orientation estimate of EKF from IMU body frame to gravity-aligned world frame, $p$ and $v$ represent the position and linear velocity estimates, and $b_g$ and $b_a$ denote the bias estimates of the gyroscope and accelerometer respectively. The error state of the EKF is propagated as a linearized error and so the error state is defined as

$$\tilde{\xi}_i = (\tilde{\theta}_i, \delta\tilde{p}_i) \quad \tilde{s} = (\tilde{\theta}, \delta\tilde{v}, \delta\tilde{p}, \delta\tilde{b_g}, \delta\tilde{b_a})$$

where the tilde indicates errors in every state. The errors for all states, except orientation, are approximated to simple subtraction even though the $SE(3)$ parameterization we use in the EKF process model is $T(3) \times SO(3)$. For orientation, we use the logarithm map of rotation to find the error as $\tilde{\theta} = log_{SO(3)}(R\hat{R}^{-1}) \in \mathfrak{so}(3)$. The EKF and error state is of dimension (6n + 15) and the corresponding state covariance matrix P has dimension (6n + 15, 6n + 15).

#### A.6.2 PROCESS MODEL

IMU's measure sequences of data $\{(a_k, \omega_k)\}_{i=k}^m$, each expressed in the local IMU frame at time $t_k$. These are related to the true IMU acceleration $\bar{a}_k$ and angular rates $\bar{\omega}_k$ via

$$\omega_k = \bar{\omega}_k + b_k^g + \eta_k^g \qquad a_k = \bar{a}_k - {}^w_b R_k^T \vec{g} + b_k^a + \eta_k^a \tag{14}$$

where $\vec{g}$ is gravity vector pointing downward in world frame, and $\eta_k^g$ and $\eta_k^a$ are IMU noises respectively. The EKF propagation uses raw IMU samples in the local IMU frame $b$, following strap-down inertial kinematics equations:

$$_b^w\hat{\mathbf{R}}_{\mathbf{k+1}} = {}_b^w\hat{\mathbf{R}}_{\mathbf{k}} \exp_{SO(3)}((\omega_{\mathbf{k}} - \hat{\mathbf{b}}_{\mathbf{gk}})\Delta t)$$

$$^w\hat{\mathbf{v}}_{\mathbf{k+1}} = {}^w\hat{\mathbf{v}}_{\mathbf{k}} + {}^w\mathbf{g}\Delta t + {}_b^w\hat{\mathbf{R}}_{\mathbf{k}}(\mathbf{a_k} - \hat{\mathbf{b}}_{\mathbf{ak}})\Delta t$$

$$^w\hat{\mathbf{p}}_{\mathbf{k+1}} = {}^w\hat{\mathbf{p}}_{\mathbf{k}} + {}^w\hat{\mathbf{v}}_{\mathbf{k}}\Delta t + \frac{1}{2}\Delta t^2({}^w\mathbf{g} + {}_b^w\hat{\mathbf{R}}_{\mathbf{k}}(\mathbf{a_k} - \hat{\mathbf{b}}_{\mathbf{a}}\mathbf{k}))$$

$$\hat{\mathbf{b}}_{\mathbf{g(k+1)}} = \hat{\mathbf{b}}_{\mathbf{gk}}$$

$$\hat{\mathbf{b}}_{\mathbf{a(k+1)}} = \hat{\mathbf{a}}_{\mathbf{gk}}$$

where at timestep $k$, $\Delta t$ is the time interval, $^w g$ is the constant gravity vector, $\eta_{gdk}$ and $\eta_{adk}$ are the IMU noises that are assumed to be normally distributed, and $\exp_{SO(3)}$ is the $SO(3)$ exponential map.

The linearized error propagation is written as:

$$\tilde{s_{k+1}} = A_{k(15,15)}^s \tilde{s}_k + B_{k(15,12)}^s n_k$$

where $n_k = [n_{\omega k}, n_{ak}, \eta_{gdk}, \eta_{adk}]$ and the subscript brackets (.,.) indicate matrix dimensions. The corresponding linearized propagation of the state covariance P is as follows:

$$P_{k+1} = A_k P_k A_k^T + B_k W B_k^T$$

$$A_k = \begin{bmatrix} I_{6n} & 0 \\ 0 & A_k^s \end{bmatrix} \quad B_k = \begin{bmatrix} 0 \\ B_k^s \end{bmatrix}$$

where $I$ stands for identity matrix amd $W_{(12,12)}$ is the covariance matrix of sensor noise and bias random walk.

### A.6.3 STATE AUGMENTATION

As the NN inputs are used to correct the EKF estimates at a frequency of 20Hz while the propagation of the EKF is at 1kHz as mentioned in TLIO, at the measurement update frequency a new state is augmented with a copy operation incrementing the dimension as seen below:

$$P_{k+1} = \bar{A}_k P_k \bar{A}_k^T + \bar{B}_k W \bar{B}_k^T$$

$$\bar{A}_k = \begin{bmatrix} I_{6n} & 0 \\ 0 & A_k^\xi \\ 0 & A_k^s \end{bmatrix} \quad \bar{B}_k = \begin{bmatrix} 0 \\ B_k^\xi \\ B_k^s \end{bmatrix}$$

where $A_k^\xi$ and $B_k^\xi$ are the partial propagation matrices for rotation and position only with dimensions (6, 15) and (6, 12) respectively.

### A.6.4 MEASUREMENT MODEL

The measurement model in the EKF uses the displacement estimates provided by the NN, aligning them in a local gravity-aligned frame to ensure the measurements are decoupled from global yaw information:

$$\hat{h}(\mathbf{X}) = \mathbf{R}_\gamma^T(\mathbf{p}_j - \mathbf{p}_i) = \hat{d}_{ij} + \eta_{ij}$$

where $\mathbf{R}_\gamma$ is the yaw rotation matrix, $\mathbf{p}_i$ and $\mathbf{p}_j$ are positions of the first and last clone state for the 1-second displacement prediction window, $\hat{d}_{ij}$ indicates the predicted displacement which is the output of the NN and $\eta_{ij}$ represents the measurement noise modeled by the network's uncertainty output as $\eta_{ij} = \mathcal{N}(0, \hat{\Sigma}_{ij})$. The orientation estimate of the EKF at clone state i is decomposed using extrinsic "XYZ" Euler angle convention as $R_i = R_\gamma R_\beta R_\alpha$.

### A.6.5 UPDATE MODEL

The Kalman gain is computed based on the measurement and covariance matrices, and the state and covariance are updated accordingly. The key update equations involve the computation of the Kalman gain ($\mathbf{K}$), updating the state ($\mathbf{X}$), and updating the covariance matrix ($\mathbf{P}$):

$$\mathbf{K} = \mathbf{P}\mathbf{H}^T(\mathbf{H}\mathbf{P}\mathbf{H}^T + \hat{\Sigma}_{ij})^{-1}$$

$$\mathbf{X} \longleftarrow \mathbf{X} \oplus \mathbf{K}(h(\mathbf{X}) - \hat{d}_{ij})$$

$$\mathbf{P} \longleftarrow (\mathbf{I} - \mathbf{K}\mathbf{H})\mathbf{P}(\mathbf{I} - \mathbf{K}\mathbf{H})^T + \mathbf{K}\hat{\Sigma}_{ij}\mathbf{K}^T$$

where $\oplus$ denotes addition operation except for rotation where the update operation writes $R \longleftarrow \exp(\tilde{\theta})R$ and the linearized measurement matrix $\mathbf{H}_{(3,6n+15)}$ which has zeros other than

$$\mathbf{H}_{\tilde{\theta}_i} = \frac{\partial h(X)}{\partial \tilde{\theta}_i} = \hat{R}_\gamma^T \lfloor {}^w\hat{p}_j - {}^w\hat{p}_i \rfloor_\times \mathbf{H}_z$$

$$\mathbf{H}_{\delta \tilde{p}_i} = \frac{\partial h(X)}{\partial \delta \tilde{p}_i} = -\hat{R}_\gamma^T$$

$$\mathbf{H}_{\delta \tilde{p}_j} = \frac{\partial h(X)}{\partial \delta \tilde{p}_j} = \hat{R}_\gamma^T$$

where

$$\mathbf{H}_z = \begin{bmatrix} 0 & 0 & 0 \\ 0 & 0 & 0 \\ \cos\gamma\tan\beta & \sin\gamma\tan\beta & 1 \end{bmatrix}$$

and $\lfloor x \rfloor_\times$ is a skew-symmetric matrix built from a vector $x$.

### A.7 EVALUATION METRICS DEFINITION

We follow most metrics in TLIO (Liu et al., 2020) and RONIN (Herath et al., 2020), besides the $MSE$ loss we reported in the paper. Here we provide the mathematical details of these metrics.

- MSE ($m^2$): Translation error per sample between the predicted and ground truth displacement averaged over the trajectory. It is computed as $\frac{1}{n}\sum_i^n \| {}^w d_i - {}^w \hat{d}_i \|^2$. However, it should be noted that MSE mentioned in TLIO (Liu et al., 2020) is the same as MSE Loss calculated as the squared error averaged separately for each axis $\frac{1}{n}\sum_i^n \| {}^w d_{i,r} - {}^w \hat{d}_{i,r} \|^2$ where $r$ is an axis.

- ATE (m): Translation Error assesses the discrepancy between predicted and ground truth (GT) positions across the entire trajectory. It is computed as $\sqrt{\frac{1}{n}\sum_i^n \| {}^w p_i - {}^w \hat{p}_i \|^2}$

- RTE (m): Relative Translation Error measures the local differences between predicted and GT positions over a specified time window of duration $\delta t$ (1 minute). $\sqrt{\frac{1}{n}\sum_i^n \| {}^w p_{i+\delta t} - {}^w p_i - ({}^w \hat{p}_{i+\delta t} - {}^w \hat{p}_i) \|^2}$.

- AYE Absolute Yaw Error is calculated as $\sqrt{\frac{1}{n}\sum_i^n \| \gamma_i - \hat{\gamma}_i \|^2}$.

### A.8 VISUALIZATION OF TLIO RESULTS

Figure 8 and Figure 9 show only the NN results compared to ground truth displacements. The ATE and RTE are calculated on the cumulative trajectory obtained from the predicted displacements. Figure 9 is with whisker extended to include the outlier which is commonly calculated as 1.5 * IQR (inter-quartile range). Figure 10 shows the results of EKF without excluding the outliers. We provide more trajectory visualizations of TLIO test data in Figure 11.

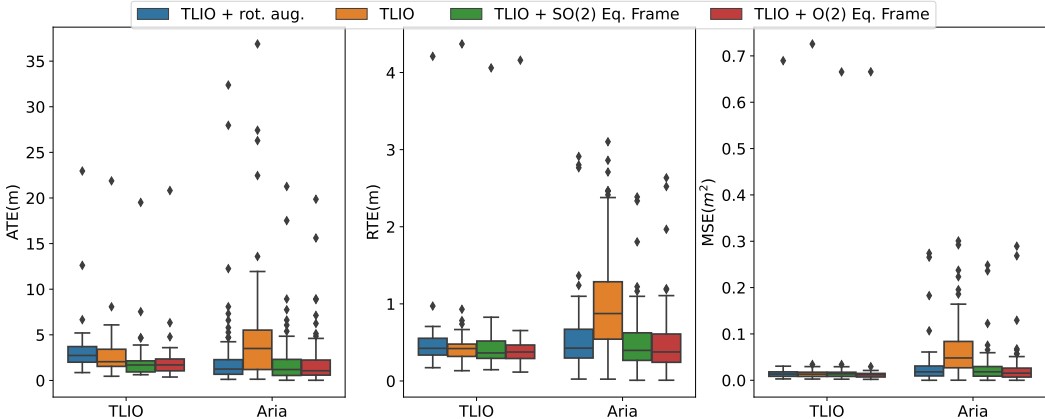

Figure 8: The superior performance of our framework applied to TLIO architecture when compared to baseline TLIO trained with and without augmentations on TLIO and Aria Datasets visualized with a box plot. Blue, Orange, Green, and Red indicate +rot. aug., TLIO, +$SO(2)$ Eq. Frame and +$O(2)$ Eq. Frame. ATE, RTE, and MSE indicate ATE*, RTE*, and MSE* corresponding to only the NN results.

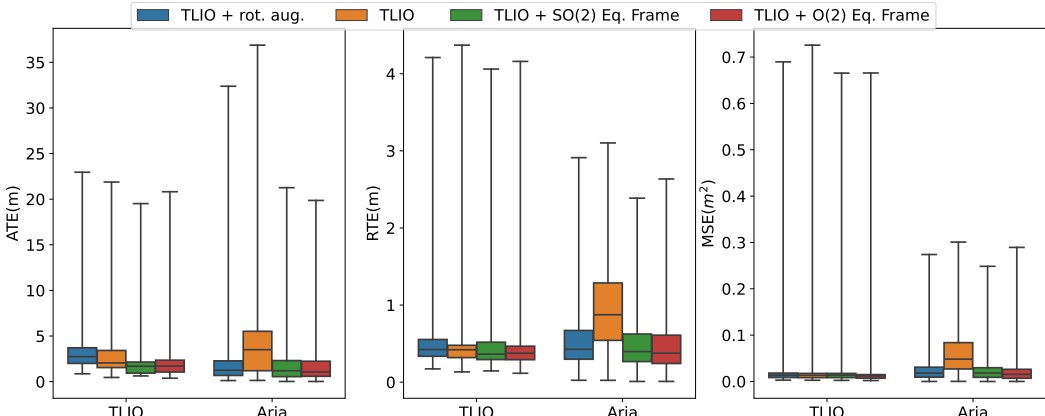

Figure 9: The superior performance of our framework applied to TLIO architecture when compared to baseline TLIO trained with and without augmentations on TLIO and Aria Datasets visualized with a box plot. Blue, Orange, Green, and Red indicate +rot. aug., TLIO, +$SO(2)$ Eq. Frame and +$O(2)$ Eq. Frame. ATE, RTE, and MSE indicate ATE*, RTE*, and MSE* corresponding to only the NN results. The whisker is extended to 1.5 * IQR (inter-quartile range).

### A.9 AUGMENTED TLIO TEST DATASET RESULTS AND ANALYSIS

We also perform an ablation study on test data augmentation for our model. For NN results, we apply four random yaw rotations per trajectory and random rotations plus reflection per trajectory. The results are detailed in Table 5. Except for our equivariant model, all other methods show decreased performance compared to their results on non-augmented test data, whereas our model maintains consistent performance and outperforms the other methods.

For the Extended Kalman Filter (EKF) results, we augment the test data using random $SO(3)$ rotations. Notably, we do not include reflections due to the structural constraints of the Kalman filter. As shown in Table 6, despite the +Non Eq. Frame model outperforming ours in non-augmented tests on ATE metrics. Our model exceeds +Non Eq. Frame on the augmented dataset. Our approach not only sets a new benchmark but also maintains consistent performance across random rotations.

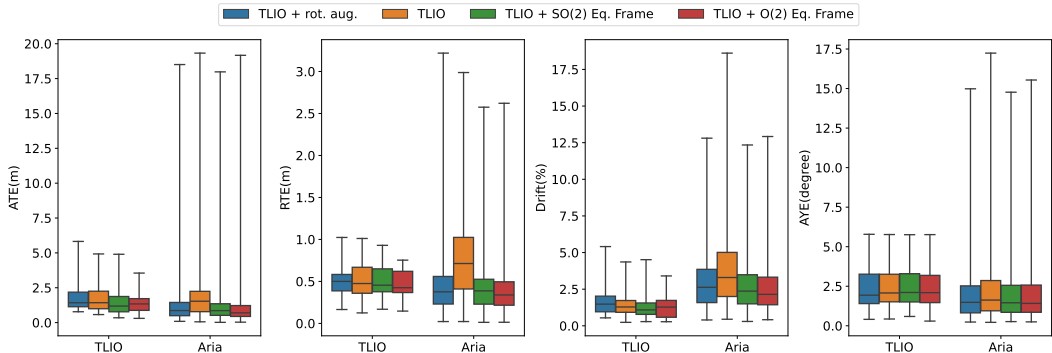

Figure 10: The superior performance of our framework applied to TLIO architecture when compared to baseline TLIO trained with and without augmentations on TLIO and Aria Datasets visualized with a box plot. Blue, Orange, Green, and Red indicate +rot. aug., TLIO, +$SO(2)$ Eq. Frame and +$O(2)$ Eq. Frame. The whisker is extended to 1.5 * IQR (inter-quartile range).

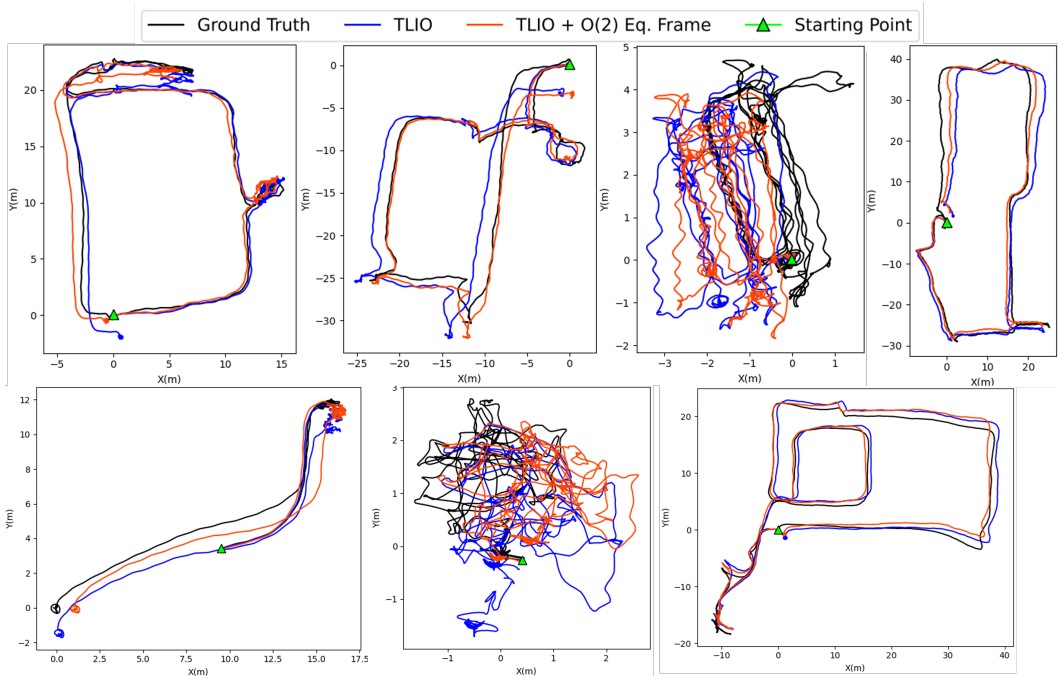

Figure 11: More Visualizations of final estimated trajectories on TLIO Dataset by baseline TLIO (Blue), our best method applied to TLIO (+$O(2)$ Eq. Frame)(Red), and the Ground-Truth trajectory (Black).

## A.10 VISUALIZATION OF RONIN

The visualization of trajectories in RONIN is displayed in Figure 12.

## A.11 COVARIANCE CONSISTENCY

Similar to TLIO (Liu et al., 2020), we plot the prediction error against the standard deviation ($\sigma$) predicted by the network in the invariant space. As seen in Figure 14 and Figure 13 the

| Model | Rotations | | | Rotations + Reflections | | |
|---|---|---|---|---|---|---|
| | MSE* $(10^{-2}m^2)$ | ATE* $(m)$ | RTE* $(m)$ | MSE* $(10^{-2}m^2)$ | ATE* $(m)$ | RTE* $(m)$ |
| TLIO | 0.2828 | 27.7797 | 3.1390 | 0.2989 | 23.4839 | 3.1313 |
| + rot. aug. | 0.0327 | 3.3180 | 0.5417 | 0.0347 | 2.9110 | 0.5654 |
| + rot. aug. + more layers | 0.0306 | 3.0264 | 0.5300 | 0.0332 | 2.3028 | 0.5592 |
| + rot. aug. + Non Eq. Frame | 0.0302 | 2.6379 | 0.5025 | 0.0331 | 2.3212 | 0.5446 |
| + rot. aug. + PCA Frame | 0.2286 | 21.3795 | 2.5288 | 0.2467 | 10.1660 | 2.2283 |
| **+ SO(2) Eq. Frame** | 0.0319 | 2.3218 | 0.4957 | 0.0339 | 1.8664 | 0.5178 |
| **+ O(2) Eq. Frame** | 0.0298 | 2.3305 | 0.4719 | 0.0298 | 1.6418 | 0.4361 |

Table 5: Ablation Study For Neural Network with Random Rotation and Reflection Transformation (4 per trajectory) on TLIO test dataset. A lower error indicates a better model. The lowest values are annotated with Red. Our proposed methods are in bold.

| Model | ATE $(m)$ | RTE $(m)$ | Drift | AYE $(deg)$ |
|---|---|---|---|---|
| TLIO | 10.3005 | 3.6263 | 2.9501 | 3.3684 |
| + rot. aug. | 1.6744 | 0.4944 | 1.5526 | 2.7290 |
| + rot. aug. + more layers | 1.6447 | 0.5466 | 1.2767 | 2.7279 |
| + rot. aug. + Non Eq. Frame | 1.4924 | 0.5119 | 1.2721 | 2.7109 |
| + rot. aug. + PCA Frame | 8.5787 | 2.9962 | 2.0872 | 3.0183 |
| **+ SO(2) Eq. Frame** | 1.4850 | 0.4901 | 1.3029 | 2.7615 |
| **+ O(2) Eq. Frame** | 1.4316 | 0.4592 | 1.3096 | 2.7250 |

Table 6: Results of evaluation of EKF with Random Rotation Transformations (4 per trajectory) on TLIO test dataset (*i.e.*, results on augmentated test dataset). A lower error indicates a better model. The lowest values are annotated with Red. Our proposed methods are in bold.

covariance prediction of our method is consistently within the $3\text{-}\sigma$ depicted by the red lines. These results show that our diagonal covariance prediction in the invariant space is consistent.

## A.12 ABLATION ON IMU SEQUENCE LENGTH

We aligned the sequence length with baseline models for fair comparison. However, in this Section, we ablate on the sequence length as shown in Table 7 and Table 8. Table 7 varies sequence lengths and displacement prediction windows (e.g., 0.5s displacement with 0.5s of 200Hz IMU data results in a sequence length of 100). Table 8 fixes the prediction window at 1 second and varies the context window (e.g., a 2s context window with 200Hz IMU data results in a sequence length of 400). Our results confirm TLIO (Liu et al., 2020) Sec. VII A.1: increasing the context window reduces MSE but not ATE. A lower MSE loss over the same displacement window does not translate to a lower ATE. Thus, the addition of the equivariant framework does not change the characteristics of the base (off-the-shelf) model used.

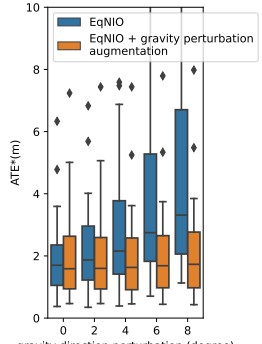

Figure 15: Sensitivity to gravity axis perturbation, and effectiveness of perturbation augmentation during training.

## A.13 SENSITIVITY ANALYSIS TO GRAVITY DIRECTION PERTURBATION

As Wang et al. (2023) we study the robustness of $SO(3)$ subequivariance to slight gravity axis perturbations in Fig. 15. We perturb the gravity axis in the test set by uniformly sampled angles $\alpha \sim \mathcal{U}(-\theta, \theta)$, where $\theta = 2°, 4°, 6°, 8°$. As Liu et al. (2020) we observe that ATE*(m) can be

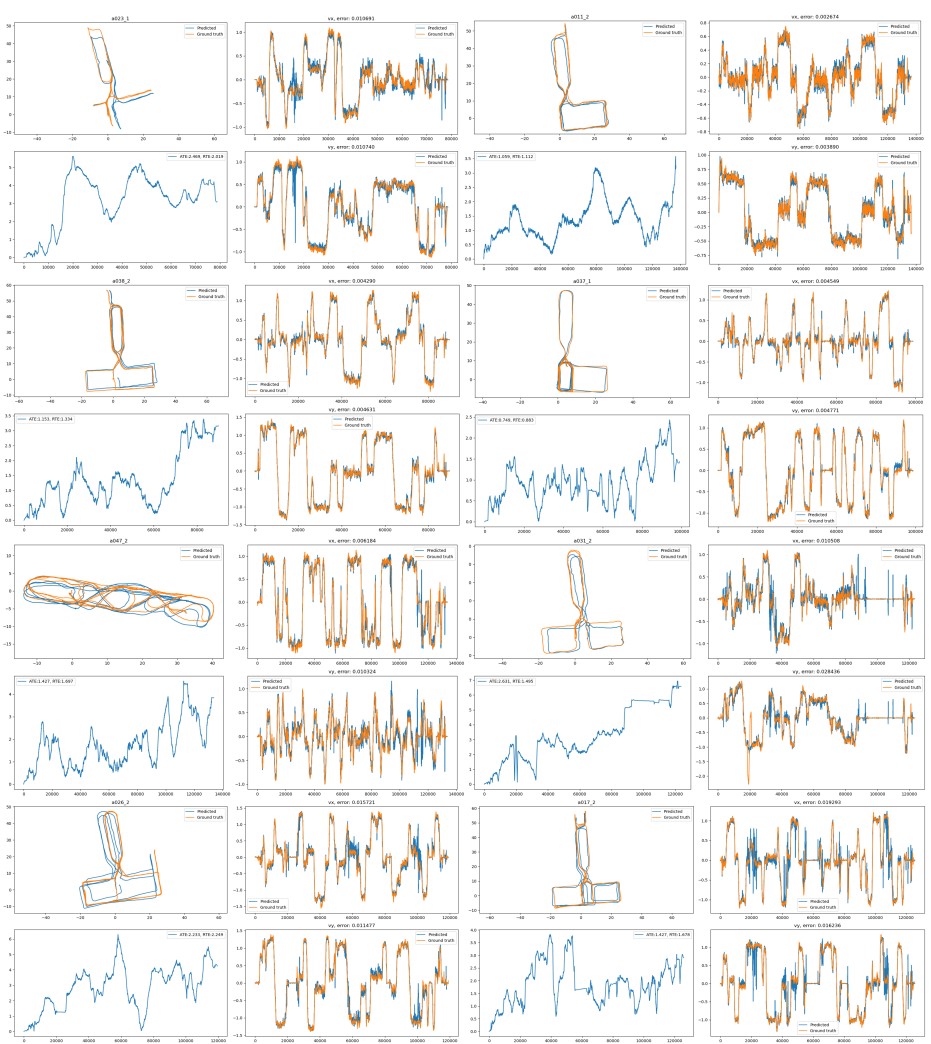

Figure 12: Visualization of RONIN Unseen Test Dataset Trajectories for our best method applied to RONIN, +$O(2)$ Eq. Frame.

stabilized by training with $\theta = 5$ gravity perturbation augmentation. The error of EqNIO trained without this augmentation increases with increasing degree of perturbation.

Table 9 presents the sensitivity analysis to gravity direction perturbation, applied for 5 different ranges, i.e., for 2°, the gravity direction perturbation of (-2°, 2°) is applied to the test dataset. We also present results for + $O(2)$ Eq. Frame model trained without the gravity direction perturbation of (-5°,5°) during training. We observe the same trend of stability in MSE* as reported in TLIO (Liu et al., 2020) when trained with gravity direction perturbation.

## A.14 SENSITIVITY TO IMU SAMPLING RATE

To study the IMU sampling rate sensitivity, we deploy our pre-trained model on IMU data that is resampled from a rate of $r \in \{50, 100, 200, 250, 500, 1000\}$ to 200 Hz, since TLIO requires a fixed input size of 200 IMU measurements for 1 second. This means that for r < 200 IMU data is interpolated. We report the results without the EKF on the Aria Dataset in Table 10.

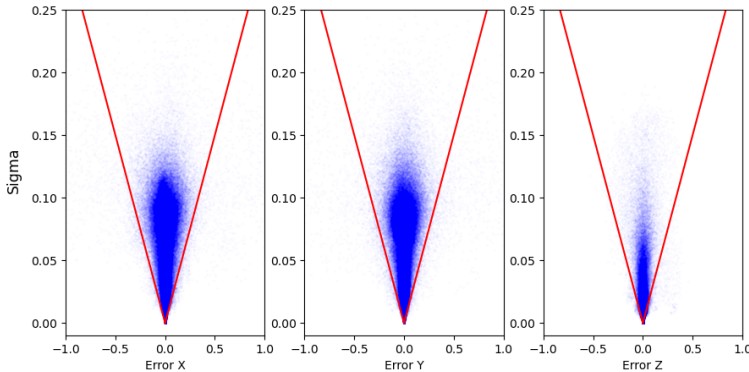

Figure 13: Consistency of Covariance Prediction in the Invariant Space for TLIO test dataset

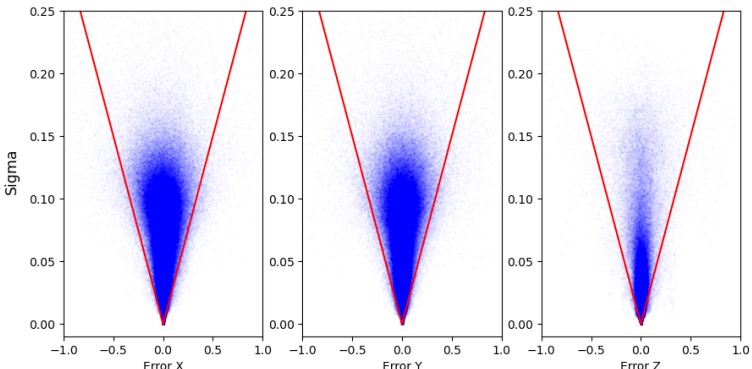

Figure 14: Consistency of Covariance Prediction in the Invariant Space for Aria dataset

The results show the relative stability of our method for rates equal to and above 200 Hz, with a maximal ATE increase of 0.2%. By contrast, going below 200 Hz leads to a higher increase of 15.8% (for 50 Hz). However, such low sampling rates are unlikely in real-world scenarios, since most commodity IMUs provide kilohertz-level sampling rates.

### A.15 SENSITIVITY TO IMU BIAS ESTIMATE

We study the sensitivity of our network to inaccurate bias estimation. For this, we monitor the MSE after perturbation of the biases used to de-bias the input IMU data to our network, as done in Liu et al. (2020). We sample uniform noise from the range $n \sim U[-r, r]$ where $r$ is defined in Table 11. Our results seen in Table 11 and Table 12 are in accordance with the sensitivity study conducted in TLIO ( Liu et al. (2020)).

### A.16 CANONICALIZATION VIA FRAME-AVERAGING

In this section, we conduct one more ablation for our frame ablation section with the frame-averaging technique in Puny et al. (2021). We first perform PCA on the IMU data, which results in a set of four frames, corresponding to the four solutions of PCA. We then use the equivariant format in Eqn.5

| Model | Context | TLIO Dataset | | | Aria Dataset | | |
|---|---|---|---|---|---|---|---|
| TLIO | Window (s) | MSE* $(10^{-2}m^2)$ | ATE* $(m)$ | RTE* $(m)$ | MSE* $(10^{-2}m^2)$ | ATE* $(m)$ | RTE* $(m)$ |
| + rot. aug. | 0.5 | 1.132 | 2.029 | 0.340 | 1.038 | 1.489 | 0.332 |
| + rot. aug. | 1 | 3.242 | 3.722 | 0.551 | 5.322 | 2.103 | 0.521 |
| + rot. aug. | 2 | 9.862 | 5.102 | 0.944 | 6.717 | 3.452 | 0.970 |
| + $SO(2)$ Eq. Frame | 0.5 | 1.124 | 0.711 | 0.175 | 1.040 | 0.673 | 0.190 |
| + **SO(2) Eq. Frame** | 1 | 3.194 | 2.401 | 0.501 | 2.457 | 1.864 | 0.484 |
| + $SO(2)$ Eq. Frame | 2 | 10.019 | 3.862 | 0.797 | 6.569 | 2.745 | 0.774 |
| + $O(2)$ Eq. Frame | 0.5 | 1.040 | 0.595 | 0.136 | 1.002 | 0.589 | 0.148 |
| + **O(2) Eq. Frame** | 1 | 2.982 | 2.382 | 0.479 | 2.304 | 1.849 | 0.465 |
| + $O(2)$ Eq. Frame | 2 | 9.804 | 4.268 | 0.762 | 6.112 | 2.556 | 0.709 |

Table 7: Results for ablation on changing prediction displacement window on TLIO architecture.

| Model | Context | TLIO Dataset | | | Aria Dataset | | |
|---|---|---|---|---|---|---|---|
| TLIO | Window ($s$) | MSE* $(10^{-2}m^2)$ | ATE* $(m)$ | RTE* $(m)$ | MSE* $(10^{-2}m^2)$ | ATE* $(m)$ | RTE* $(m)$ |
| + rot. aug. | 1 | 3.242 | 3.722 | 0.551 | 5.322 | 2.103 | 0.521 |
| + rot. aug. | 2 | 3.199 | 2.555 | 0.511 | 3.790 | 2.895 | 0.713 |
| + rot. aug. | 3 | 3.284 | 4.463 | 0.617 | 3.511 | 3.014 | 0.738 |
| + **SO(2) Eq. Frame** | 1 | 3.194 | 2.401 | 0.501 | 2.457 | 1.864 | 0.484 |
| + $SO(2)$ Eq. Frame | 2 | 2.886 | 1.837 | 0.429 | 2.187 | 1.533 | 0.444 |
| +$SO(2)$ Eq. Frame | 3 | 2.790 | 3.090 | 0.492 | 1.986 | 1.684 | 0.447 |
| + **O(2) Eq. Frame** | 1 | 2.982 | 2.382 | 0.479 | 2.304 | 1.849 | 0.465 |
| + $O(2)$ Eq. Frame | 2 | 2.382 | 1.895 | 0.367 | 1.307 | 1.382 | 0.338 |
| + $O(2)$ Eq. Frame | 3 | 2.161 | 2.083 | 0.366 | 0.974 | 1.672 | 0.366 |

Table 8: Results for ablation on changing context window with fixed displacement window of 1-second on TLIO architecture.

of Puny et al. (2021) which canonicalizes the input with respect to the four frames, runs a separate prediction on each input, and then averages the outputs. Note that this scales the inference time by a factor of 4. We show the results in Table 13. We see that while TLIO + frame averaging achieves a low error on the TLIO dataset it fails to generalize to the Aria Dataset. Furthermore, our equivariant method (TLIO + O(2) Eq. Frame) outperforms it on both datasets and across metrics. We believe that the subpar performance of TLIO + frame averaging stems from the noise sensitivity of PCA. Due to this sensitivity, it likely overfits to the specific noise level present in the TLIO dataset, which does not match the one in the Aria dataset.

### A.17 ANALYSIS OF THE COVARIANCE PARAMETERIZATIONS

In addition to supporting our design choice, of a diagonal covariance parameterization in the canonical frame empirically by running our NN (see Table 3), we add additional statistical analyses below, using two more techniques: *(i)* we report the percentage of displacement predictions by our network that have an error outside of the 3-sigma bound, in x,y, and z direction (denoted $\delta_x, \delta_y, \delta_z$ in percent), where sigma is given by our network, and *(ii)* we compute the median negative log-likelihood (median NLL) of the ground truth samples given the mean and covariance provided by our method. All results were calculated in the canonical frame and pertain to the Aria Dataset. In addition, we highlight the importance of learning the covariance, by introducing two additional baselines: First, similar to Liu et al. (2020), we use our pre-trained method to generate displacement measurements but use a constant isotropic covariance ($\sigma = 0.01$) instead of the learned one (indicated with +constant cov), and secondly we train our model only on MSE for 20 epochs (indicated with (mse)+constant cov) and deploy it with the same constant covariance. We see in Table 14 that TLIO + O(2) Eq. Frame, TLIO + O(2) Eq. Frame + P and TLIO + O(2) Eq. Frame (mse) + constant cov, all show low outlier

| Model | $\vec{g}$ direction perturbation | TLIO Dataset | | |
|---|---|---|---|---|
| (TLIO) | (deg) | MSE* $(10^{-2}m^2)$ | ATE* $(m)$ | RTE* $(m)$ |
| **+ SO(2) Eq. Frame** | 0 | 3.194 | 2.401 | 0.501 |
| + $SO(2)$ Eq. Frame | 2 | 3.201 | 2.409 | 0.500 |
| + $SO(2)$ Eq. Frame | 4 | 3.206 | 2.404 | 0.498 |
| + $SO(2)$ Eq. Frame | 6 | 3.241 | 2.442 | 0.501 |
| + $SO(2)$ Eq. Frame | 8 | 3.298 | 2.502 | 0.506 |
| + $O(2)$ Eq. Frame ‡ | 0 | 2.982 | 2.406 | 0.478 |
| + $O(2)$ Eq. Frame ‡ | 2 | 3.198 | 2.663 | 0.498 |
| + $O(2)$ Eq. Frame ‡ | 4 | 3.742 | 3.292 | 0.559 |
| + $O(2)$ Eq. Frame ‡ | 6 | 4.505 | 4.228 | 0.659 |
| + $O(2)$ Eq. Frame ‡ | 8 | 5.433 | 5.218 | 0.768 |
| **+ O(2) Eq. Frame** | 0 | 2.981 | 2.382 | 0.479 |
| + $O(2)$ Eq. Frame | 2 | 2.988 | 2.390 | 0.480 |
| + $O(2)$ Eq. Frame | 4 | 3.010 | 2.415 | 0.486 |
| + $O(2)$ Eq. Frame | 6 | 3.060 | 2.471 | 0.486 |
| + $O(2)$ Eq. Frame | 8 | 3.095 | 2.506 | 0.489 |

Table 9: Results for ablation on changing prediction displacement window on TLIO +$O(2)$ Eq. Frame. ‡ implies the network was trained without gravity direction perturbation.

| Model | Sampling rate | MSE* | ATE* | RTE* |
|---|---|---|---|---|
| (TLIO) | (Hz) | $(10^{-2}m^2)$ | $(m)$ | $(m)$ |
| + $O(2)$ Eq. Frame | 1000 | 2.324 | 1.876 | 0.470 |
| + $O(2)$ Eq. Frame | 500 | 2.316 | 1.878 | 0.469 |
| + $O(2)$ Eq. Frame | 250 | 2.291 | 1.882 | 0.468 |
| + $O(2)$ Eq. Frame | 200 | 2.276 | 1.881 | 0.468 |
| + $O(2)$ Eq. Frame | 100 | 2.191 | 1.932 | 0.470 |
| + $O(2)$ Eq. Frame | 50 | 2.256 | 2.174 | 0.501 |

Table 10: Results for ablation on changing IMU sampling rate on TLIO + $O(2)$ Eq. Frame. on Aria Dataset

counts. However, low outlier counts can also be achieved by predicting high covariances. However, this strategy increases the median NLL, as seen in TLIO + O(2) Eq. Frame + P and TLIO + O(2) Eq. Frame (mse) + constant cov, showing that they overestimate covariances. Our method (TLIO + O(2) Eq. Frame) shows the highest median NLL overall methods, with low outlier counts. For completeness, we also report the tracking performance of the new methods in Table 15. We see that, first, O(2) Eq. Frame + constant cov performs worse than our method, indicating that our learned covariance adapts to the specific learned displacements. Second, we see that (mse)+constant cov performs even worse. This is likely due to the network being overconfident in its displacement prediction before MLE finetuning, which results in significant outlier prediction.

| Model (TLIO) | Accel Bias Range $(m/s^2)$ | MSE* $(10^{-2}m^2)$ |
|---|---|---|
| + $O(2)$ Eq. Frame | 0.0 | 2.981 |
| + $O(2)$ Eq. Frame | 0.1 | 2.984 |
| + $O(2)$ Eq. Frame | 0.2 | 3.001 |
| + $O(2)$ Eq. Frame | 0.3 | 3.031 |
| + $O(2)$ Eq. Frame | 0.4 | 3.066 |
| + $O(2)$ Eq. Frame | 0.5 | 3.110 |

Table 11: Results for ablation on accelerometer bias sensitivity on TLIO + $O(2)$ Eq. Frame. on TLIO test Dataset

| Model (TLIO) | Gyro Bias Range $(rad/s)$ | MSE* $(10^{-2}m^2)$ |
|---|---|---|
| + $O(2)$ Eq. Frame | 0.000 | 2.981 |
| + $O(2)$ Eq. Frame | 0.025 | 2.981 |
| + $O(2)$ Eq. Frame | 0.050 | 2.981 |
| + $O(2)$ Eq. Frame | 0.075 | 2.981 |
| + $O(2)$ Eq. Frame | 0.100 | 2.981 |

Table 12: Results for ablation on gyroscope bias sensitivity on TLIO + $O(2)$ Eq. Frame. on TLIO test Dataset

| Model | TLIO Dataset | | | Aria Dataset | | |
|---|---|---|---|---|---|---|
| | MSE* $(10^{-2}m^2)$ | ATE* $(m)$ | RTE* $(m)$ | MSE* $(10^{-2}m^2)$ | ATE* $(m)$ | RTE* $(m)$ |
| TLIO | | | | | | |
| + rot. aug. | 3.242 | 3.722 | 0.551 | 5.322 | 2.103 | 0.521 |
| + **SO(2) Eq. Frame** | 3.194 | 2.401 | 0.501 | 2.457 | 1.864 | 0.484 |
| + **O(2) Eq. Frame** | 2.982 | 2.382 | 0.479 | 2.304 | 1.849 | 0.465 |
| + frame averaging | 3.208 | 3.057 | 0.536 | 5.821 | 4.554 | 0.992 |

Table 13: Results for the frame-averaging technique evaluating only the neural network performance on the TLIO and Aria Datasets.

| Model (TLIO) | $\delta_x$ (%) | $\delta_y$ (%) | $\delta_z$ (%) | Median NLL |
|---|---|---|---|---|
| + $O(2)$ Eq. Frame +S | 3.49 | 3.35 | 7.75 | -7.0551 |
| + **O(2) Eq. Frame** | 0.59 | 0.71 | 0.51 | -7.3191 |
| + $O(2)$ Eq. Frame +P | 0.00 | 0.96 | 0.76 | -4.1156 |
| + $O(2)$ Eq. Frame + constant cov | 2.18 | 2.20 | 0.46 | -4.1111 |
| + $O(2)$ Eq. Frame (mse) + constant cov | 0.72 | 0.72 | 0.04 | -3.9189 |

Table 14: Analysis of covariance parameterizations by measuring the number of outliers outside of the 3-$\sigma$ bound, and the negative log-likelihood (NLL) of ground truth samples on the Aria Dataset.

| Model (TLIO) | TLIO Dataset | | | Aria Dataset | | |
|---|---|---|---|---|---|---|
| | ATE $(m)$ | RTE $(m)$ | AYE $(deg)$ | ATE $(m)$ | RTE $(m)$ | AYE $(deg)$ |
| + $O(2)$ Eq. Frame +S | 1.484 | 0.462 | 2.390 | 1.175 | 0.421 | 2.043 |
| + **O(2) Eq. Frame** | 1.433 | 0.458 | 2.389 | 1.118 | 0.416 | 2.059 |
| + $O(2)$ Eq. Frame +P | 1.827 | 0.578 | 2.534 | 1.755 | 0.564 | 2.223 |
| + $O(2)$ Eq. Frame + constant cov | 1.669 | 0.506 | 2.481 | 1.680 | 0.534 | 2.197 |
| + $O(2)$ Eq. Frame (mse) + constant cov | 2.883 | 0.799 | 2.477 | 2.032 | 0.652 | 2.224 |

Table 15: EKF Results for the different covariance parameterizations

