# OpenReview forum: "EqNIO: Subequivariant Neural Inertial Odometry"
_ICLR.cc/2025/Conference — ICLR 2025 Poster_

### Official Review · Reviewer_EZfo · 2024-11-04

**Soundness:** 3
**Presentation:** 3
**Contribution:** 3
**Rating:** 8
**Confidence:** 4

**Summary:**

This paper  presents a method to enhance inertial odometry by applying group equivariance to canonicalize IMU data and targeting yaw ambiguity in gravity-aligned frames through a subequivariant framework.

The authors design a neural network architecture that maintains equivariance under roto-reflections around the gravity axis, allowing integration with existing systems like RONIN and TLIO. By predicting canonical yaw frames and equivariant covariance matrices, EqNIO improves generalization across diverse motion patterns and reduces drift caused by sensor noise and biases. Experiments on publicly available datasets demonstrate that this method achieves reductions in Mean Squared Error and Absolute Translation Error compared to baseline models, while also exhibiting faster convergence and maintaining computational efficiency suitable for deployment on edge devices.

**Strengths:**

The paper introduces an approach by applying strict subequivariance in neural inertial odometry, addressing yaw ambiguity and gravity alignment limitations directly within the network architecture, an aspect that prior methods often handled indirectly. Its originality lies in developing a canonicalization scheme using the roto-reflection group to simplify IMU data processing. The authors integrate and extend existing methods by adapting the framework to both end-to-end and filter-based neural inertial odometry systems, which demonstrates its flexibility and scalability.

In terms of quality, the paper provides detailed, reproducible implementation notes and thorough ablation studies in the appendix that clarify design decisions and evaluate parameterization choices. It emphasizes empirical rigor by testing the model across multiple datasets with varied sensor placements and motion patterns, supporting claims of robustness and broad applicability. Clarity is maintained through structured explanations of complex mathematical formulations, specifically around group theory and its relevance to sensor data processing, while the significant computational efficiency results underscore the practical utility for edge-device applications. The combination of technical insights and comprehensive empirical validation underlines the paper's contribution to advancing neural inertial odometry, particularly in settings with challenging orientations and device constraints.

**Weaknesses:**

The abstract describes EqNIO as leveraging "canonical displacement priors" to generalize across arbitrary IMU orientations, but it lacks a clear technical explanation of how these priors work in practice. Generalization is claimed to stem from "canonical gravity-aligned frames" and "equivariant yaw frames," but the abstract could benefit from a more precise explanation of these transformations and their operationalization in the model.

The learnable yaw orientation in canonical frames is a promising feature but lacks clarity on how it resolves yaw drift or improves orientation estimation, given that yaw is typically the most challenging to estimate accurately in inertial odometry due to the absence of an absolute reference.

The introduction highlights EqNIO’s generalization and robustness but does not discuss potential limitations or scenarios where the approach may struggle (e.g., handling different sampling rates, extreme motions where IMU biases may not be fully mitigated, or contexts with poor gravity alignment).

While EqNIO is compared to existing neural odometry methods like TLIO and RONIN, the introduction does not delve into specific weaknesses in these prior approaches and how EqNIO addresses these limitations.

The paper covers a broad range of related works but may omit some recent or seminal papers in the domain of learning-based inertial odometry and equivariant neural networks. Ensure a comprehensive literature review by including all relevant and recent works that contribute to the field. This includes verifying that seminal papers and the latest advancements are adequately cited to position EqNIO within the current research landscape. Due to the inherent relationship between odometry and inertial attitude estimation, as well as the similar methods applied to both, I highly encourage you to explore these areas further, including learning-based approaches to inertial attitude estimation.

The descriptions of related methods (e.g., TLIO, RONIN) are somewhat high-level and lack technical depth.  Providing only superficial descriptions may not adequately highlight the nuances that differentiate EqNIO from these methods.

**Questions:**

Could you provide a technical explanation of how the canonical displacement priors are implemented in practice?
How exactly do the gravity-aligned frames and equivariant yaw frames work in your model architecture?
What specific mechanisms in your learnable yaw orientation approach help address the yaw drift problem?
Could you provide experimental evidence demonstrating how your method improves yaw estimation compared to existing approaches?
How does your model perform under varying IMU sampling rates?
What are the performance characteristics under extreme motion scenarios where IMU biases may be significant?
How does the system behave in situations with poor gravity alignment?
How does your work relate to recent developments in learning-based inertial attitude estimation?
Could you elaborate on the connections between EqNIO and current research in equivariant neural networks specifically applied to inertial navigation?

---

> ### Author Response · Authors · 2024-11-24
>
> **W1. The abstract describes EqNIO as leveraging "canonical displacement priors" to generalize across arbitrary IMU orientations, but it lacks a clear technical explanation of how these priors work in practice. Generalization is claimed to stem from "canonical gravity-aligned frames" and "equivariant yaw frames," but the abstract could benefit from a more precise explanation of these transformations and their operationalization in the model.**
>
> Thanks for the valuable suggestion. What we and previous work refer to as “neural displacement priors” are neural networks that take IMU measurements as input and generate displacement and covariance predictions as outputs, i.e. a prior probability distribution on the expected displacement, given a set of measurements. These priors can be incorporated into probabilistic filters to estimate the state of the IMU over time. When referring to “canonical priors”, we mean priors (i.e. predicted displacement and covariances) expressed in a canonical frame. We uniquely define the canonical frame via three axes: the first is the gravity axis (making the frame gravity aligned), and the second and third are predicted by an “equivariant frame model” that takes IMU data as input. Note that this frame is not necessarily right-handed, and quantities expressed in this frame can, therefore, potentially undergo a reflection and rotation. This model is termed “equivariant” since it commutes with rotation around or reflection across the gravity axis (O(2) roto-reflection). This means that if the input is O(2) roto-reflected by an arbitrary transformation, the resulting output is roto-reflected by the same transformation.
>
> We reworked the abstract to lay more emphasis on the explanations above and provide further details on the **operationalization** of the canonical frames in Q1 below.
>
> **W2. The learnable yaw orientation in canonical frames is a promising feature but lacks clarity on how it resolves yaw drift or improves orientation estimation, given that yaw is typically the most challenging to estimate accurately in inertial odometry due to the absence of an absolute reference.**
>
> We believe that the term “learnable yaw” has generated some confusion. The canonical frame is specified by gravity and two learnable orthogonal vectors perpendicular to gravity. It does not define a yaw measurement and is thus not incorporated directly into the filter. Instead, this frame is used to canonicalize the input, generate the displacement, and then transform it back into the original frame. As a result, our network generates more consistent outputs (see Fig. 1) when data is observed under different O(2) roto-reflections within the plane perpendicular to gravity. Since the network does not need to learn to generate consistent results across O(2) roto-reflections, it can overall generate better displacement and covariance estimates and, thus, improve the trajectory tracking in the EKF (when the frame is applied to TLIO). Since the measurement equation of the EKF depends on the IMU yaw orientation (see Eq. 14 in Liu et al. 2020, and Appendix A.6.4), better displacement measurements should result in better IMU yaw estimation and thus lower yaw drift. However, experimentally we observed this effect is very small, showing only a small improvement on the Aria Dataset (Tab. 3 see 2.073 deg average yaw error (AYE) for TLIO + rot. aug. vs. 2.059 deg AYE for TLIO + O(2) Eq. Frame).

---

> > ### Author Response · Authors · 2024-11-24
> >
> > **W3. The introduction highlights EqNIO’s generalization and robustness but does not discuss potential limitations or scenarios where the approach may struggle (e.g., handling different sampling rates, extreme motions where IMU biases may not be fully mitigated, or contexts with poor gravity alignment).**
> >
> > We highlight the sensitivity of our method with respect to the following parameters: Poor gravity alignment, a study which we already report in Fig. 7 of the original version, has been moved to Appendix A.13 Fig. 15 for consistency and space constraints;  and, following the suggestions, we provide the following additional sensitivity studies on the sampling rate, and IMU bias accuracy in the Appendices A.14 and A.15.
> >
> > To study IMU sampling rate sensitivity, we deploy our pre-trained model on IMU data that is resampled from a rate of  r $\in$ \{50, 100, 200, 250, 500, 1000\} to 200 Hz, since TLIO requires a fixed input size of 200 IMU measurements for 1-second. This means that for r < 200 IMU data is interpolated. We report the results without the EKF in the loop below:
> >
> >
> > | Sampling Rate | Aria MSE*  | Aria ATE*  | Aria RTE*  |
> > |---------------|------------|------------|------------|
> > | 1000          | 0.023242   | 1.876466   | 0.470085   |
> > | 500           | 0.023164   | 1.878051   | 0.469828   |
> > | 250           | 0.022908   | 1.882291   | 0.468969   |
> > | 200           | 0.022763   | 1.881573   | 0.468524   |
> > | 100           | 0.021909   | 1.931893   | 0.470235   |
> > | 50            | 0.022564   | 2.174362   | 0.500758   |
> >
> > This shows the relative stability of our method for rates equal to and above 200 Hz, with a maximal ATE increase of 0.2%. By contrast, going below 200 Hz leads to a higher increase of 15.8% (for 50 Hz). However, such low sampling rates are unlikely in real-world scenarios, since most commodity IMUs provide kilohertz-level sampling rates.
> >
> > Next, we study the sensitivity of our network to inaccurate bias estimation. For this, we monitor the MSE after perturbation of the biases used to de-bias the input IMU data to our network, as done by Liu et al. 2020. We sample uniform noise from the range $\nu\sim U[-r, r]$ where $r$ is defined in the tables below.
> >
> > | Gyro Bias Range | MSE*   |
> > |-----------------|--------|
> > | 0.000           | 0.02981 |
> > | 0.025           | 0.02981 |
> > | 0.050           | 0.02981 |
> > | 0.075           | 0.02981 |
> > | 0.100           | 0.02981 |
> >
> >
> > | Accel Bias Range | MSE*   |
> > |------------------|--------|
> > | 0.0              | 0.02981 |
> > | 0.1              | 0.02984 |
> > | 0.2              | 0.03001 |
> > | 0.3              | 0.03031 |
> > | 0.4              | 0.03066 |
> > | 0.5              | 0.03110 |
> >
> > Our results are in accordance with the sensitivity study conducted in TLIO (Liu et al. (2020)).
> >
> > We include a section discussing these sensitivity studies in Appendices A.12, A.13, A.14, and A.14 of the revision.
> >
> > **W4. While EqNIO is compared to existing neural odometry methods like TLIO and RONIN, the introduction does not delve into specific weaknesses in these prior approaches and how EqNIO addresses these limitations.**
> >
> > Our work addresses the following three weaknesses of prior work: First, prior networks are only trained to be consistent with limited types of IMU data transformations: Liu et al. 2020 (TLIO), Herath et al. 2020 (RONIN), and Cao et al. 2022 (RIO)  are only trained to be consistent when applying **rotations** around the gravity axis. In our work, we train our model to be consistent when applying **roto-reflections**, and this requires the development of a novel bijection for preprocessing angular rates. We see that modeling this additional transformation improves results consistently. Secondly, RIO and RONIN only target trajectory tracking in 2D, while our method can flexibly track in 2D or 3D. This is because the network outputs of RIO and RONIN are only 2D velocities in the xy-plane. Our method can handle the equivariance transformations of 3D displacements and 3x3 covariances. Finally, TLIO, RONIN, and RIO all employ approximate equivariance to ensure consistency under the above transformations. This means that they either use data augmentation, auxiliary equivariant consistency losses or Test-Time-Training to minimize the inconsistency between network outputs from data under different rotations around gravity. Our method guarantees this inconsistency to be 0, throughout training and testing, by employing equivariant neural network layers that produce consistent outputs **by design**. We show in Tabs. 1 and 2, the resulting networks consistently outperform all previous works by large margins.

---

> > > ### Author Response · Authors · 2024-11-24
> > >
> > > **W5. The paper covers a broad range of related works but may omit some recent or seminal papers in the domain of learning-based inertial odometry and equivariant neural networks. Ensure a comprehensive literature review by including all relevant and recent works that contribute to the field. This includes verifying that seminal papers and the latest advancements are adequately cited to position EqNIO within the current research landscape. Due to the inherent relationship between odometry and inertial attitude estimation, as well as the similar methods applied to both, I highly encourage you to explore these areas further, including learning-based approaches to inertial attitude estimation.**
> > >
> > > We thank the reviewer for the valuable feedback. In the revision Section 2 and Appendix A.1, we supplement our related work section with the following works in equivariant odometry without learning and inertial attitude estimation. We further invite the reviewer to suggest more.
> > >
> > > We added a series of works [1,3,4,5]  about embedding equivariance or symmetry into (visual) inertial odometry, but these methods are not learning-based. As far as we know, we are the first to bake the equivariance into the neural network for the inertial odometry problem. Integrating our equivariant network into these models would be a potential future work.  They work on equivariant dynamics and use the lift function to map the state and the extended velocity to the Lie algebra associated with the system's symmetry group, enabling the construction of a lifted system on the Lie group. The error dynamics are linearized around a fixed origin, which makes it independent of the current state.
> > >
> > > We added recent related works [2,6,7,8,9] on inertial odometry and inertial attitude estimation. In particular, inertial attitude estimation works do not address equivariance, and extending their work in this way is a potential avenue for future research. A particular difference to our setting is that they do not require gravity alignment, and thus the formulation would deviate slightly from the one in this work. The work in [7] extends the EKF filter to be a learnable component, which is an orthogonal research direction to ours, and can further enhance the performance of our method.
> > > We added the initial works on EKFs for inertial attitude estimation [10,11,12] in Appendix A.1 and also the most recent learning-based methods for LiDAR Odometry [13], VIO [14] and SLAM [15].
> > >
> > > **Please refer to the following box for the references**

---

> > > > ### Author Response · Authors · 2024-11-24
> > > >
> > > > **W5 references**
> > > >
> > > > [1] Equivariant IMU Preintegration with Biases: an Inhomogeneous Galilean Group Approach
> > > >      Giulio Delama, Alessandro Fornasier, Robert Mahony, Stephan Weiss
> > > >      https://arxiv.org/abs/2411.05548
> > > >
> > > > [2] Geometric Data Fusion for Collaborative Attitude Estimation
> > > >     Yixiao Ge, Behzad Zamani, Pieter van Goor, Jochen Trumpf , Robert Mahony
> > > >     https://www.sciencedirect.com/science/article/pii/S2405896324019566
> > > >
> > > > [3] MSCEqF: A Multi State Constraint Equivariant Filter for Vision-Aided Inertial Navigation
> > > >      Alessandro Fornasier,  Pieter van Goor, Eren Allak, Robert Mahony, Stephan Weiss
> > > >     https://ieeexplore.ieee.org/abstract/document/10325586
> > > >
> > > > [4]  Equivariant Symmetries for Inertial Navigation Systems
> > > > Alessandro Fornasier, Yixiao Ge, Pieter van Goor, Robert Mahony, Stephan Weiss
> > > > https://arxiv.org/abs/2309.03765
> > > >
> > > > [5] EqVIO: An Equivariant Filter for Visual-Inertial Odometry
> > > > Pieter van Goor, Robert Mahony
> > > > https://ieeexplore.ieee.org/abstract/document/10179117
> > > >
> > > > [6] Generalizable End-to-End Deep Learning Frameworks for Real-Time Attitude Estimation Using 6DoF Inertial Measurement Units
> > > > Arman Asgharpoor Golroudbari, Mohammad Hossein Sabour
> > > > https://arxiv.org/pdf/2302.06037
> > > >
> > > > [7] AirIMU: Learning Uncertainty Propagation for Inertial Odometry
> > > > Yuheng Qiu, Chen Wang, Can Xu, Yutian Chen, Xunfei Zhou, Youjie Xia, Sebastian Scherer
> > > > https://arxiv.org/pdf/2310.04874
> > > >
> > > > [8] Initialization techniques for 3D SLAM: A survey on rotation estimation and its use in pose graph optimization
> > > >  Luca Carlone, Roberto Tron, Kostas Daniilidis and Frank Dellaert
> > > > https://www.cis.upenn.edu/~kostas/mypub.dir/tron15icra.pdf
> > > >
> > > > [9] RIANN—A Robust Neural Network Outperforms Attitude Estimation Filters
> > > >      Daniel Weber, Clemens Gühmann, Thomas Seel
> > > >  https://arxiv.org/pdf/2104.07391
> > > >
> > > > [10] Attitude estimation for UAV using extended Kalman filter
> > > >      Xiaofei Jing, Jiarui Cui, Hongtai He, Bo Zhang, Dawei Ding, Yue Yang
> > > >     https://ieeexplore.ieee.org/document/7979077
> > > >
> > > > [11] Quaternion-Based Robust Attitude Estimation Using an Adaptive Unscented Kalman Filter
> > > >       Chiella ACB, Teixeira BOS, Pereira GAS
> > > >      https://www.mdpi.com/1424-8220/19/10/2372
> > > >
> > > > [12] Quaternion attitude estimation for miniature air vehicles using a multiplicative extended Kalman filter
> > > > James K. Hall, Nathan B. Knoebel, Timothy W. McLain
> > > > https://ieeexplore.ieee.org/document/4570043
> > > >
> > > > [13] Tightly-Coupled LiDAR-IMU-Wheel Odometry with Online Calibration of a Kinematic Model for Skid-Steering Robots
> > > > Taku Okawara, Kenji Koide, Shuji Oishi, Masashi Yokozuka, Atsuhiko Banno, Kentaro Uno, Kazuya Yoshida
> > > > https://ieeexplore.ieee.org/stamp/stamp.jsp?arnumber=10681089
> > > >
> > > > [14] Adaptive VIO: Deep Visual-Inertial Odometry with Online Continual Learning
> > > > Youqi Pan, Wugen Zhou, Yingdian Cao, Hongbin Zha
> > > > https://openaccess.thecvf.com/content/CVPR2024/papers/Pan_Adaptive_VIO_Deep_Visual-Inertial_Odometry_with_Online_Continual_Learning_CVPR_2024_paper.pdf
> > > >
> > > > [15] DVI-SLAM: A Dual Visual Inertial SLAM Network
> > > > Xiongfeng Peng, Zhihua Liu, Weiming Li, Ping Tan, SoonYong Cho, Qiang Wang
> > > > https://arxiv.org/abs/2309.13814

---

> > > > > ### Author Response · Authors · 2024-11-24
> > > > >
> > > > > **W6. The descriptions of related methods (e.g., TLIO, RONIN) are somewhat high-level and lack technical depth. Providing only superficial descriptions may not adequately highlight the nuances that differentiate EqNIO from these methods.**
> > > > >
> > > > > Thank you for the suggestion.  We have a list of differences between our method and the related methods, as well as include more details of related methods in the revision Appendix A.4, where changes are denoted in red. For specific weaknesses of these approaches that are addressed by our method, please refer to W4.
> > > > >
> > > > > **Q1. Could you provide a technical explanation of how the canonical displacement priors are implemented in practice?**
> > > > >
> > > > > Please find a technical explanation below, which we incorporated into Appendix A.5.1.
> > > > >
> > > > > (1) Following Fig. 2 (a) we first gravity align the IMU data, i.e. rotate it such that the z-axis is aligned with gravity (estimated online by an EKF)
> > > > >
> > > > >
> > > > > (2) We then process this IMU data by an “equivariant frame model” that outputs two vectors. This network takes in vector and scalar features derived from the IMU data. The equivariant network is strictly O(2) or SO(2) equivariant by design. The predicted vectors are then converted into an orthogonal set of unit vectors using Gram-Schmidt orthogonalization.
> > > > >
> > > > > (3) Then, we transform the input IMU data using the equivariant frame from (2) (“canon.” block in Fig. 2(a)) to produce invariant (consistent) inputs within the canonical frame, ensuring robustness to any roto-reflection group transformations applied to the original data.
> > > > >
> > > > > (4) The invariant input is fed into a standard neural network architecture (“off-the-shelf model”), such as TLIO's ResNet, to generate a displacement and covariance, termed canonical displacement prior expressed with respect to the learned canonical frame. These outputs remain consistent under transformations from the roto-reflection group applied to the original input.
> > > > >
> > > > > (5) Lastly, we project back the predicted displacement and covariance using the canonical frame to obtain an equivariant displacement and covariances using the equivariant frame of (2).
> > > > >
> > > > > Finally, depending on the backbone architecture on which our framework is applied (in our case for TLIO),  the predicted displacement and covariances are fed into a filtering algorithm, like an EKF as a measurement to update the estimate of the IMU state (orientation, position, velocity and biases).
> > > > >
> > > > > **Q2. How exactly do the gravity-aligned frames and equivariant yaw frames work in your model architecture?**
> > > > >
> > > > > For each 1-second window of IMU data, there is exactly one gravity-aligned frame and one equivariant yaw frame, outlined next:
> > > > >
> > > > > The gravity-aligned frame is the frame into which the IMU data is transformed before processing by the equivariant frame model. This frame has its z-axis aligned with gravity but is otherwise unconstrained. We map the IMU data into this frame by simply rotating it along the shortest path such that the z-axis points toward gravity. We use the gravity direction estimated by the EKF for this step.
> > > > >
> > > > > The equivariant yaw frame is defined as a composition of the gravity-aligned frame with a roto-reflection around gravity. It is defined by three axes: the z-axis being the gravity, and the other two being provided by the equivariant frame model. This frame is not necessarily right-handed. After producing these two vectors, we express the IMU data in this new frame, which we call the canonical frame. We then run the neural network and produce an invariant covariance and displacement (i.e. a canonical displacement prior). We then project the outputs of this network using this yaw frame into the original gravity-aligned frame. This entire process enhances the generalization of the network due to the reasons discussed in W1. For more details of the implementation of the canonical frame please refer to Q1.
> > > > >
> > > > > **Q3.  What specific mechanisms in your learnable yaw orientation approach help address the yaw drift problem?**
> > > > >
> > > > > See W4.
> > > > >
> > > > > **Q4. Could you provide experimental evidence demonstrating how your method improves yaw estimation compared to existing approaches?**
> > > > >
> > > > > As mentioned previously in W2, our method produces more accurate displacement measurements than previous approaches, highlighted by the reduced ATE Tables 2 and 3. This improvement in displacement measurements should imply an improvement in yaw estimation due to the measurement equation’s dependence on the EKF yaw state (see Appendix A.6.4, A.6.5 for the Jacobian of the measurement with respect to the yaw). However, in practice, we only find a small improvement in terms of average yaw error on Aria (Tab. 3 see 2.073 deg average yaw error (AYE) for TLIO + rot. aug. vs. 2.059 deg AYE for TLIO + O(2) Eq. Frame). Since the Aria dataset is out of distribution with respect to the TLIO training set, this also highlights the slightly superior generalization ability of our method.
> > > > >
> > > > > **Q5.How does your model perform under varying IMU sampling rates?**
> > > > >
> > > > > See W3.

---

> > > > > > ### Author Response · Authors · 2024-11-24
> > > > > >
> > > > > > **Q6. What are the performance characteristics under extreme motion scenarios where IMU biases may be significant?**
> > > > > >
> > > > > > See W3.
> > > > > >
> > > > > > **Q7. How does the system behave in situations with poor gravity alignment?**
> > > > > >
> > > > > > See W3.
> > > > > >
> > > > > > **Q8. How does your work relate to recent developments in learning-based inertial attitude estimation?**
> > > > > >
> > > > > > See W5.
> > > > > >
> > > > > > **Q9. Could you elaborate on the connections between EqNIO and current research in equivariant neural networks specifically applied to inertial navigation?**
> > > > > >
> > > > > > See W5.

---

> > > > > > > ### Author Response · Authors · 2024-11-30
> > > > > > >
> > > > > > > Respected Reviewer, we greatly appreciate your feedback and comments and have addressed them in our preceding responses. We look forward to your feedback.

---

### Official Review · Reviewer_AwiV · 2024-11-04

**Soundness:** 3
**Presentation:** 3
**Contribution:** 3
**Rating:** 8
**Confidence:** 2

**Summary:**

This paper presents a new method for inertial odometry, which predicts the poses given IMU measurements. The method is called EqNIO, which brings the idea of the so-called canonical displacement priors. How it works, is (1) that IMU measurements are mapped into a gravity aligned, canonical frame with neural networks, and (2) mapping the outputs back to the original frame. Several contributions are presented. A canonicalization scheme is presented, that maps IMU measurements into a canonical orientation. A processing step is devised, which map both accelerometer and gyro readings into a space where gravity direction is preserved. Finally, a neural network designed is presented to perform regression tasks. Several experiments are presented, demonstrating advancements to the state of the art and ablation studies that motivate the overall approach.

**Strengths:**

Pros:

-	The paper is written well with clear figures, and many intuitive explanations of complex concepts.

-	The paper presents several technical contributions, which overall leads to more generalizable framework for inertial odometry. Mixing the physical properties with learning based regression module makes sense, which can boost generalization performance.

-	Experimental results are presented to a large extent, demonstrating advancements to the state of the art. Ablation studies presented are useful to better comprehend the research done.

**Weaknesses:**

Cons:

-	It is not clear if ICLR is the best venue for such research, since learning components here is rather limited to a regression module.

-	Uncertainty modelling assumes diagonal covariance. Validity of these assumptions are tested by looking at the final performance that it helps. Perhaps an in-depth analysis on this step could help, despite not the core focus of the paper. For example, there has been many evaluation tools from uncertainty quantification literature, and can be presented here.

**Questions:**

I wonder for sensor fusion in the form of visual-inertial odometry, it could be helpful to have a good uncertainty estimates from the inertial odometry module. Uncertainty estimates could consider the modelling errors of neural networks, and propagated to the final module. Here, priors can also be defined using physical properties of the system. Would it be a consideration to use Bayesian modelling tools?

---

> ### Author Response · Authors · 2024-11-24
>
> **W1. It is not clear if ICLR is the best venue for such research, since learning components here is rather limited to a regression module.**
>
>  Equivariance has been a prevalent topic in ICLR. Because the canonicalization is a preprocessing step and yields a group action, the data transformed with this action can be fed into any architecture beyond regression.
>
> **W2. Uncertainty modelling assumes diagonal covariance. Validity of these assumptions are tested by looking at the final performance that it helps. Perhaps an in-depth analysis on this step could help, despite not the core focus of the paper. For example, there has been many evaluation tools from uncertainty quantification literature, and can be presented here.**
>
> We apologize for not formulating this in a clear way. The covariance is diagonal in the canonical frame but after back-transforming it with a roto-reflection (predicted by the canonicalization) the resulting covariance becomes non-diagonal. This is a feature of our system that the meaning of our canonical frames is the orientation of the covariance matrix.
>
> In addition to supporting this design choice empirically by running our neural network (see Tab. 3), we add additional statistical analyses below, using two more techniques: (1) we report the percentage of displacement predictions by our network that have an error outside of the 3-sigma bound, in x,y, and z direction (denoted $\delta_x,\delta_y,\delta_z$ in percent), where sigma is given by our network, and (2) we compute the median negative log-likelihood (median NLL) of the ground truth samples given the mean and covariance provided by our method. All results were calculated in the canonical frame and pertain to the Aria Dataset.
>
> In addition, we highlight the importance of learning the covariance, by introducing two additional baselines: First, similar to Liu et al 2020, we use our pre-trained method to generate displacement measurements, but use a constant isotropic covariance (sigma=0.01) instead of the learned one (indicated with +constant cov), and secondly we train our model only on MSE for 20 epochs (indicated with (mse)+constant cov) and deploy it with the same constant covariance.
>
>
> | TLIO  					| $\delta_x$  | $\delta_y$ | $\delta_z$ |  median NLL |
> | ------- | ----------------- | ----- | ----- | ----- |
> | + O(2) Eq. Frame +S                      		| 3.49 | 3.35 | 7.75 | -7.0551 |
> | + O(2) Eq. Frame                                    	| 0.59 | 0.71 | 0.51 | -7.3191 |
> | + O(2) Eq. Frame +P                              	| 0.00 | 0.96 | 0.76 | -4.1156 |
> | + O(2) Eq. Frame + constant cov           	| 2.18 | 2.20 | 0.46 | -4.1111 |
> | + O(2) Eq. Frame (mse) + constant cov 	| 0.72 | 0.72 | 0.04 | -3.9189 |
>
> We see that TLIO + O(2) Eq. Frame, TLIO + O(2) Eq. Frame + P and TLIO + O(2) Eq. Frame (mse) + constant cov, all show low outlier counts. However, low outlier counts can also be achieved by predicting high covariances. However, this strategy increases the median NLL, as seen in TLIO + O(2) Eq. Frame + P and TLIO + O(2) Eq. Frame (mse) + constant cov, showing that they overestimate covariances. Our method (TLIO + O(2) Eq. Frame) shows the highest median NLL overall methods, with low outlier counts.
>
> For completeness, we also report the tracking performance of the new methods below
> | TLIO                                 | TLIO ATE  | TLIO RTE  | TLIO AYE  | Aria Avg ATE | Aria Avg RTE | Aria Avg AYE |
> |--------------------------------------|-----------|-----------|-----------|--------------|--------------|--------------|
> | + O(2) Eq. Frame +S                  | 1.4836    | 0.4623    | 2.3902    | 1.1752       | 0.4211       | 2.0433       |
> | + O(2) Eq. Frame                     | 1.4328    | 0.4583    | 2.3894    | 1.1181       | 0.4159       | 2.0592       |
> | + O(2) Eq. Frame +P                  | 1.8267    | 0.5776    | 2.5342    | 1.7546       | 0.5636       | 2.2234       |
> | + O(2) Eq. Frame + constant cov      | 1.6691    | 0.5063    | 2.4811    | 1.6801       | 0.5335       | 2.1971       |
> | + O(2) Eq. Frame (mse) + constant cov | 2.8827    | 0.7988    | 2.4769    | 2.0319       | 0.6521       | 2.2244       |
>
> We see that, first, O(2) Eq. Frame + constant cov performs worse than our method, indicating that our learned covariance adapts to the specific learned displacements. Second, we see that (mse)+constant cov performs even worse. This is likely due to the network being overconfident in its displacement prediction before MLE finetuning, which results in significant outlier prediction.
>
> We have included this analysis in Appendix A.17 of the revision. We look forward to your suggestions to further analyze the various covariance parameterizations used in the paper.

---

> > ### Author Response · Authors · 2024-11-24
> >
> > **Q1.  I wonder for sensor fusion in the form of visual-inertial odometry, it could be helpful to have a good uncertainty estimate from the inertial odometry module. Uncertainty estimates could consider the modeling errors of neural networks, and propagated to the final module. Here, priors can also be defined using the physical properties of the system. Would it be a consideration to use Bayesian modelling tools?**
> >
> > Thanks for providing the new direction and potential fusion of our model, these are indeed excellent questions and directions for research.
> > First, we think integrating our work into visual odometry would be a direct future work following the procedure of (Chen et al., 2021a), which uses the provided displacement and uncertainties in a VIO backend.
> > Second, our current EKF is already an instance of Bayesian filtering with a Gaussian noise assumption, where model prediction is the displacement prior, and the measurement step maximizes the posterior. Nonetheless, we believe that using additional Bayesian modeling of the displacement noise, leveraging specific physical properties of the system, such as moment of inertia, or sensor temperature, would greatly benefit our method. Leveraging more sophisticated filters like the Unscented Kalman Filter or Particle Filter would additionally benefit our method. Finally, we believe that future directions should incorporate not only aleatoric uncertainties (about data), which we used in the current work but also epistemic uncertainties that arise from the model.

---

> > > ### Author Response · Authors · 2024-11-30
> > >
> > > Respected Reviewer, we greatly appreciate your feedback and comments and have addressed them in our preceding responses. We look forward to your feedback.

---

### Official Review · Reviewer_wVti · 2024-11-08

**Soundness:** 2
**Presentation:** 2
**Contribution:** 3
**Rating:** 5
**Confidence:** 5

**Summary:**

This work introduces EqNIO, a neural network-based odometry system that enhances localization accuracy using accelerometer and gyroscope data from a single IMU. Traditional neural odometry methods face challenges with generalization, as variations in IMU orientation and motion direction can disrupt displacement predictions. EqNIO overcomes this by training a model with canonical displacement priors, aligning IMU data to a gravity-aligned frame with learnable yaw. This approach ensures that the system’s outputs are invariant to rotations and reflections in the gravity direction, supporting robust generalization. Through carefully designed layers and an innovative angular rate decomposition, EqNIO can effectively integrate both acceleration and angular data. Tested on TLIO, Aria, RONIN, RIDI, and OxIOD datasets, EqNIO demonstrates superior performance and adaptability over existing methods, marking a step forward in low-drift neural inertial odometry suitable for edge devices.

**Strengths:**

I think the method itself looks novel and interesting.  It introduces a canonicalization scheme that leverages gravity and an estimated sub-equivariant frame to map IMU measurements into a canonical orientation. This procedure can be flexibly applied to arbitrary off-the-shelf network architectures by mapping the inputs into the canonical space and mapping the outputs back into the original space.

**Weaknesses:**

The primary weakness of this paper is the clarity of its writing. I’m unable to fully understand the major differences between this work and RIO: Rotation-equivariance Supervised Learning of Robust Inertial Odometry.

While the key idea of this paper is clear, it’s difficult to discern how it specifically diverges from the previous work. I strongly recommend that the authors begin by clearly outlining the main concepts, followed by a detailed description of the methodology. This structure would greatly help readers in understanding the unique contributions of this work.

**Questions:**

What is the roto-reflection group, and why is it important? A more detailed explanation of this concept and its relevance would be helpful.

What is the PCA(handcrafted equivariant frame)? A more detailed explanation of this concept and its relevance would be helpful.

Clarity in distinguishing from RIO: It appears that the figure is intended to convey the core idea of this work. However, the differences between this approach and RIO are unclear—elaborating on this distinction would strengthen the presentation.

Data specification in figure captions: It would be beneficial if each figure caption specified which data is seen and which is unseen to enhance the reader's understanding. Note, the performance of different method highly depends on how much data is seen or trained.

Supplementary material vs. main paper clarity: The supplementary material provides much better clarity than the main paper. Including some of this contextual information directly in the main paper would make it easier for reviewers to follow your method.

---

> ### Author Response · Authors · 2024-11-24
>
> **W1. The primary weakness of this paper is the clarity of its writing. I’m unable to fully understand the major differences between this work and RIO: Rotation-equivariance Supervised Learning of Robust Inertial Odometry.**
>
>  While our work and RIO both aim to address the equivariance of neural inertial odometry by modeling rigid transformations of trajectory and IMU data, there are several distinct differences.
>
> (1) Most significantly, RIO addresses **approximate** equivariance, while our method enforces **strict equivariance**. Approximate equivariance enforces equivariance via an equivariance loss which penalizes inconsistencies in predictions on data from four rotated trajectories. By contrast, **strict**  equivariance enforces **inherent** consistency by tailoring specific neural network components to be **exactly** equivariant. This work proposes several specialized linear and non-linear layers to guarantee the strict equivariance of the predicted canonical frame, while RIO uses traditional layers.
>
> (2) RIO adopts a Test-Time Training strategy, which adds computational overhead and thus increases the inference time as they are required to store multiple versions of the prediction model and augmented trajectories. Since our method is strictly equivariant, we do not require a Test-Time Training strategy. Moreover, despite not using such a strategy our method outperforms RIO.
>
> (3) RIO only handles $2D$ displacement outputs and equivariance to rotations in SO(2), i.e. $2D$ planar rotations. By contrast, our method can handle both $2D$ and $3D$ outputs and thus addresses equivariance in $2D$ and subequivariance in $3D$.
>
> (4) Last, we also extend the modeling to roto-reflections, i.e. the group $O(2)$, which comprises rotations and reflections in the plane perpendicular to gravity and requires a novel bijection for angular rate preprocessing.
>
> To highlight the difference in equivariance types (strict/approximate), we included a visualization of the different strategies in the revision Appendix A.4.
>
> **W2. While the key idea of this paper is clear, it’s difficult to discern how it specifically diverges from the previous work. I strongly recommend that the authors begin by clearly outlining the main concepts, followed by a detailed description of the methodology. This structure would greatly help readers in understanding the unique contributions of this work.**
>
> As discussed previously, our work significantly diverges from the previous works mentioned above (RIO) in four major ways (i) approximate vs. strict equivariance, (ii) test-time training vs. no test-time training, and (iii) $2D$ vs. $3D$ modeling and (iv) equivariance to rotations in $SO(2)$ vs. roto-reflections $O(2)$. Please see our previous response for a recap.
>
> We also thank the reviewer for the recommendation of reworking our method, and have tried our best to include the preliminary theory that is necessary to understand the concepts in this work in Appendix A.4. We supplemented this with additional pointers to textbooks and prior works. Regretfully, we believe, due to space considerations, that it is out of scope for us to include more preliminary theory in the main text.
>
> **Q1. What is the roto-reflection group, and why is it important? A more detailed explanation of this concept and its relevance would be helpful.**
>
> The Roto-reflection group $O(n)$  is a set of $n\times n$ matrices $R$ (in this work $n=2$), which are orthogonal, i.e. $RR^T = I_n$. Their determinant can be 1 or -1. This set of matrices forms a group because (i) it has an identity element $I_n$, (ii) the product of any two matrices from $O(n)$ is in $O(n)$, and (iii) every matrix $R\in O(n)$ has an inverse $R^T\in O(n)$. As shown in Figure 3a,  the orange trajectory is obtained by reflecting the reference trajectory, and the purple one is obtained by rotating the reference trajectory. The reflections and rotations are the transformations in the roto-reflection group. We illustrate how this transformation acts on the inputs of the neural network in Figure 3b.

---

> > ### Author Response · Authors · 2024-11-24
> >
> > **Q2. What is the PCA(handcrafted equivariant frame)? A more detailed explanation of this concept and its relevance would be helpful.**
> >
> > PCA refers to the method of finding the principal axes, i.e. axes which exhibit the highest and lowest variance in the $a_{xy}$, $v_{1,xy}$ and $v_{2,xy}$ components of the $n$ IMU measurements, stacked into a  $(3n)\times 2$ matrix $X$. When a yaw transformation acts on the data, $X$ transforms as $XR^T$. PCA performs a Singular Value Decomposition (SVD) on $X = U\Sigma V^T$, where $U$, $V$ are orthogonal, $V$ is a $2\times 2$ matrix. In this case, we can use V. V  is an equivariant frame because under yaw transformations $XR^T$ we obtain $U\Sigma V^T R^T$, so $V$ becomes $RV$ after transformation of the data, i.e. it transforms equivariantly under yaw transformation. We say that such an equivariant frame is handcrafted since it is not learned from a data set. The reason why we do not use PCA is that the PCA decomposition is ambiguous up to the sign of each principal axis (eigenvector). This means that small noise perturbations can cause discontinuous changes in the frame. For this reason, canonicalizing with PCA frames results in poorer performance as compared to our method.
> >
> > **Q3. Clarity in distinguishing from RIO: It appears that the figure is intended to convey the core  idea of this work. However, the differences between this approach and RIO are unclear—elaborating on this distinction would strengthen the presentation.**
> >
> > See W1.
> >
> > **Q4.Data specification in figure captions: It would be beneficial if each figure caption specified which data is seen and which is unseen to enhance the reader's understanding. Note, the performance of different method highly depends on how much data is seen or trained.**
> >
> > The confusion is our fault and we corrected it in the revision Section 5, Table 2, and Appendix A.3. Seen and unseen labels refer to specific subsets of the RONIN test dataset. Specifically, RONIN-U (Unseen) is a test set that contains IMU measurements from people who did not participate in the training and validation data collection. The people used to record the RONIN-S (Seen) overlap with those from the training and validation set, but their data is disjoint from the training and validation set. The RONIN-U dataset, thus, tests the generalization capabilities of a method.
> >
> > To the point of “which data is seen and unseen”, we believe the reviewer is referring to the training, validation (seen), and test (unseen) datasets in each case, which we gladly added to the captions. For each table, the listed datasets were not seen by any method as they constitute the test set. For training, Tab. 1 and 3, and Figs. 4-7 each method uses the TLIO Dataset, and for Tab. 2 each method uses 50% of the RONIN training set (since only 50% is public), and we include the original RONIN result using 100% of the training data.
> >
> > We added these clarifying details in the revision Section 5, Table 2, and Appendix A.3.
> >
> > **Q5. Supplementary material vs. main paper clarity: The supplementary material provides much better clarity than the main paper. Including some of this contextual information directly in the main paper would make it easier for reviewers to follow your method.**
> >
> > We appreciate your positive feedback on the supplementary material and we have now enriched our manuscript with more contextual information and details in the revision (all changes are indicated in red) while also staying within the page limit.

---

> > > ### Author Response · Authors · 2024-11-30
> > >
> > > Respected Reviewer, we greatly appreciate your feedback and comments and have addressed them in our preceding responses. We look forward to your feedback.

---

> > > > ### Author Response · Authors · 2024-12-01
> > > >
> > > > Thank you for taking the time to consider and review our rebuttal. We would be happy to address any other questions you may have about improving our work further and provide any necessary clarifications that would result in an increase in your rating.

---

### Official Review · Reviewer_cuLt · 2024-11-09

**Soundness:** 3
**Presentation:** 3
**Contribution:** 2
**Rating:** 6
**Confidence:** 4

**Summary:**

The authors propose a method to adapt existing inertial odometry (IO) architectures to be invariant to the IMU orientation. This is done by making use of an $O_g(3)$/$SO_g(3)$ equivariant network that transforms the gravity-aligned IMU measurements to a canonical frame as a pre-processing step for non-equivariant IO. The predicted displacement and covariance from IO for these canonicalized measurements are then transformed back to the source frame using the inverse canonical frame. The proposed method leads to improved accuracy while maintaining comparable runtime and can in principle be applied to any IO method.

**Strengths:**

- This is an interesting and novel application of equivariant networks to an under-explored but useful problem. The canonicalization approach is a good choice for this problem as it adapts existing sota IO architectures and keeps the pipeline interpretable.
- The symmetry of the problem in terms of $O_g(3)$ equivariance (which is $O(3)$ subequivariant) is well presented. Care has been taken to consistently process the IMU measurements and specialized $O(2)$/$SO(2)$ architectures based on vector neurons have been developed, while more architectures are possible.
- The approach was tested on two IO architectures which showed accuracy improvements on many datasets, with comprehensive ablation studies, while keeping the inference times comparable.

**Weaknesses:**

- The general canonicalization scheme has been proposed before by Kaba, Sékou-Oumar, et al. "Equivariance with learned canonicalization functions." ICML 2023, and cannot not be presented as a contribution. The original work must be cited.
- While there is a reduction in drift compared to the baselines, the remaining drift is still significant (>2m) which suggests that the main problem in IO is not in exact equivariance but elsewhere (most likely sensor noise).

**Questions:**

- Why is this canonical. equiv. scheme chosen over other equiv. choices? e.g. frame averaging (Puny et al. ICLR '22) also allows adapting existing non-equiv. architectures.
- I'm confused about the choice of metrics, especially for the TLIO experiments. From the definitions in A.5 (I believe squared norm is missing), it seems that MSE is just sqrt(ATE)? But the numbers don't reflect this. And I also think ATE, RTE, AYE would be sufficient. Do you do SE3 alignment with the GT trajectories?

Minor non-critical comments:
- Could you elaborate on the yaw augmentation procedure used for TLIO / RoNIN?
- It is surprising to me that despite requiring 10x more FLOPs than the non-equiv. architectures, there is barely any increase in runtime (<1 ms). Since there is no code release, can you comment more on the reasons for this efficiency?
- Writing: In Fig. 3b it is not clear what 'rot. sense' means; explain how the frame is constructed from the network outputs with gs-orth. for sake of clarity; Typo in conclusion: "respects eliminates"; Would be helpful to indicate that * means no-EKF in the table 2,3 captions or simply remove the * since it is not applicable to RoNIN.

---

> ### Author Response · Authors · 2024-11-23
>
> **W1. The general canonicalization scheme has been proposed before by Kaba, Sékou-Oumar, et al. "Equivariance with learned canonicalization functions." ICML 2023, and cannot not be presented as a contribution. The original work must be cited.**
>
> Thanks for pointing out the original work, which we cited in the revision (Section 2). As correctly pointed out, we achieve equivariance via the canonicalization scheme presented in Kaba et al. We want to highlight that our novelty is in characterizing and addressing the subequivariant structure in neural inertial odometry where the learned network predicts the displacement priors. To this end, we design a subequivariant framework by **applying** canonicalization and designing specific basic layers.
>
> We made the following modifications in the revision: (i) cite the work of Kaba et al. in the related work section, and add a separate related work section on canonicalization in Appendix A.1 due to space constraints, including citations of previous and subsequent papers on canonicalization. and (ii) modify our introduction to make clear that our novelty is in **applying** canonicalization to IMU data, and that, to achieve this, we tailor specific basic equivariant layers and a bijective map.
>
> **W2. While there is a reduction in drift compared to the baselines, the remaining drift is still significant (>2m) which suggests that the main problem in IO is not in exact equivariance but elsewhere (most likely sensor noise).**
>
> The lack of principled subequivariance modeling in previous approaches accounts for a significant increase in drift, and we address this with the presented work. In fact, by correctly accounting for this geometric constraint, EqNIO reduces the MSE* by 57%, the ATE* by 12%, and the RTE* by 11% on the Aria Dataset (see Tab. 1).
>
> As rightly pointed out, there still exists drift compared to VIO, and this highlights the fact that pure IO is a historically hard, but equally exciting problem to study. Incorrect biases in addition to sensor noise are two known drivers of drift in learning-based IO. To address this, Brossard et al., 2020a, Brossard et al., 2020b and Buchanan et al., 2023 train models to predict either IMU biases or debiased IMU data directly, which reduces drift by a significant margin. We believe that our framework can readily incorporate such methods to replace the current way of debiasing IMU measurements, which relies on factory-calibrated bias values.
>
> **Q1. Why is this canonical. equiv. scheme chosen over other equiv. choices? e.g. frame averaging (Puny et al. ICLR '22) also allows adapting existing non-equiv. Architectures.**
>
> We thank the reviewer for this suggestion and present results for an additional baseline following the frame-averaging technique in Puny et al. ICLR ‘22 applied to our O(2) model. We first perform PCA (using the torch.pca_low_rank() implementation) on the IMU data, which results in a set of four frames, corresponding to the four solutions of PCA. We then use the equivariant format in Eqn.5 of Puny et al. ICLR ‘22 to average the projected predictions. We show the results below, marked as TLIO + frame averaging, and include them in Appendix A.16 of the revision:
>
> | Model                  | TLIO Dataset MSE* | TLIO Dataset ATE* | TLIO Dataset RTE* | Aria Dataset MSE* | Aria Dataset ATE* | Aria Dataset RTE* |
> |------------------------|--------------------|--------------------|--------------------|-------------------|-------------------|-------------------|
> | TLIO                  | 0.0333            | 3.0786            | 0.5418            | 0.1525            | 4.5599            | 0.9771            |
> | + rot. aug            | 0.0324            | 3.7219            | 0.5513            | 0.0532            | 2.1027            | 0.5208            |
> | + SO(2) Eq. Frame     | 0.0319            | 2.4009            | 0.5006            | 0.0246            | 1.8639            | 0.4836            |
> | + O(2) Eq. Frame      | 0.0298            | 2.4056            | 0.4775            | 0.0230            | 1.8491            | 0.4649            |
> | + frame averaging     | 0.0321            | 3.0566            | 0.5358            | 0.0582            | 4.5535            | 0.9922            |
>
> We see that while TLIO + frame averaging achieves a low error on the TLIO dataset it fails to generalize to the Aria Dataset. Furthermore, our equivariant method (TLIO + O(2) Eq. Frame) outperforms it on both datasets and across metrics. We believe that the subpar performance of TLIO + frame averaging stems from the noise sensitivity of PCA. Due to this sensitivity, it likely overfits to the specific noise level present in the TLIO dataset, which does not match the one in the Aria dataset. Lastly, we want to note that the frame averaging technique significantly increases the inference time by a factor of four, as the model needs to be inferred several times, once for every constructed frame.

---

> > ### Author Response · Authors · 2024-11-23
> >
> > **Q2. I'm confused about the choice of metrics, especially for the TLIO experiments. From the definitions in A.5 (I believe the squared norm is missing), it seems that MSE is just sqrt(ATE)? But the numbers don't reflect this. And I also think ATE, RTE, AYE would be sufficient. Do you do SE3 alignment with the GT trajectories?**
> >
> > Thanks for pointing out the typo. Here, the MSE is, as stated in the text, the mean of the square error of displacement predictions $\hat{d}_i$ over 1s-window, and not the position $\hat{p}_i$. Therefore the definition of the error should be $mean_i ||d_i -\hat{d}_i||^2$, where $||.||$ is the Euclidean norm. We have corrected this in the revision Appendix A.7.
> >
> > **Minor non-critical comments:**
> >
> > **M1. Could you elaborate on the yaw augmentation procedure used for TLIO / RoNIN?**
> >
> >  We follow the procedure in Liu et al., 2020, which randomly rotates the data associated with 1s of IMU data and the corresponding ground truth displacement around the yaw direction of the local gravity-aligned frame (z-axis, corresponding with gravity) during training. The yaw angle is sampled from the range $[-\pi, \pi]$.
> >
> > **M2. It is surprising to me that despite requiring 10x more FLOPs than the non-equiv. architectures, there is barely any increase in runtime (<1 ms). Since there is no code release, can you comment more on the reasons for this efficiency?**
> >
> > We apologize for this confusion and have noticed that there was an error in timing. The corrected runtimes are as follows: TLIO takes 2.79 ms, TLIO + SO(2) Eq. Frame takes 5.43 ms and TLIO + O(2) Eq. Frame takes 5.70 ms per inference. We incorporated this change in Appendix A.5.2 in our revision and we will release the code upon acceptance.
> >
> > **M3. Writing: In Fig. 3b it is not clear what 'rot. sense' means;**
> >
> > Rot. sense stands for the sense of rotation, i.e., the direction of spinning implied by the direction of the arrow, which follows the right-hand rule.
> >
> > **explain how the frame is constructed from the network outputs with gs-orth. for sake of clarity;**
> >
> > We predict equivariant vector features with two channels, which can be interpreted as two vectors $v_1$ and $v_2$. The gs-orth. module operates in three steps following Gram-Schmidt orthogonalization: First, it normalizes $v_1$ resulting in $f_1$. Then it subtracts the component of $v_2$ which is parallel to $v_1$ (i.e. $v_2^*=v_2 - <f_1,v_2>f_1$). Finally, it normalizes $v_2^*$ resulting in $f_2$. The frame $F = [f_1, f_2]$ is the desired equivariant, orthogonal frame.
> >
> > **Typo in conclusion: "respects eliminates";**
> >
> > The correct sentence should read: “Our canonicalization scheme eliminates the underlying yaw ambiguity in gravity-aligned frames which arise from roto-reflections in the plane around gravity.” We have updated this in revision Section 7.
> >
> > **Would be helpful to indicate that * means no-EKF in the Table 2,3 captions or simply remove the * since it is not applicable to RoNIN.**
> >
> > Thank you for the suggestions and we fixed this in the revision Tables 2 and 3 to consistently indicate no-EKF with *.

---

> ### Comment · Reviewer_cuLt · 2024-11-25
>
> Thank you for addressing all my questions and comments.
>
> The accuracy improvements still seem marginal to me (ATE / RTE are more informative performance metrics than the MSE loss) compared to the increased complexity and run-time (on-device run-time will be even higher). Nonetheless, this work is still an interesting and novel application of learned canonicalization that is well developed and has potential for future work. Thus I have increased my score from 5 to 6.

---

> > ### Author Response · Authors · 2024-11-25
> >
> > Thank you very much for your valuable feedback and review.

---

### Author Response · Authors · 2024-11-24
**Global Comment**

We thank all reviewers for their insightful and constructive feedback, and helpful comments for improving the clarity of the paper. Below we will first summarize the changes towards a clearer exposition of our work, followed by a rebuttal given to individual reviewers based on their feedback. Finally, we provide the revision of the paper with incorporated changes marked in red.

We made the following modifications to the manuscript: In the related work section, we cited the work of Kaba et al., which first proposed to achieve equivariance by learning canonicalization. We added a section on canonicalization in Appendix A.1 as suggested by Reviewer cuLt, and added more references on inertial attitude estimation and the broader fields of visual-inertial odometry and SLAM based on the feedback from Reviewer EZfo. As suggested by Reviewer cuLt, we added another ablation comparing our method to frame-averaging (Omri Puny et al., ICLR 2022), which further highlights the benefits of learning the equivariant frame. Following the suggestion by Reviewer wVti, we added a section in Appendix A.4 to provide a detailed comparison of our method with previous works like RIO, TLIO, and RONIN highlighting our main difference, which is exact equivariance as opposed to approximate equivariance. A major difference in our work is the handling of reflection symmetries in addition to simple rotation symmetries around the gravity axis (termed roto-reflections). In particular, we show that reflecting quantities across planes parallel to gravity is a viable symmetry transformation of the input and output, and designing an equivariant network to deal with this symmetry leads to improved generalization of displacement priors. As proposed by Reviewer AwiV, we conduct a more detailed uncertainty analysis of the three different covariance parameterizations explored in our paper. To this end, we quantify the proportion of 3-sigma outliers, and the negative log-likelihood of the test set under predicted means and uncertainties, and include this in the revision Appendix A.17.  As recommended by Reviewer EZfo, we conducted a sensitivity analysis by varying the IMU sampling rates and perturbing the IMU biases, and this analysis highlights the robustness of our method.

Lastly, we thank the reviewers for recognizing the novelty of EqNIO, the thoroughness of the mathematical treatment, and the impact of addressing this problem on the growing field of inertial odometry. In particular, Reviewer cuLt praises our work as *“an interesting and novel application of equivariant networks to an under-explored but useful problem”*. Reviewers wVti and cuLt recognize the novelty of our method and consider *“the canonicalization approach a good choice for this problem as it adapts existing sota IO architectures and keeps the pipeline interpretable”*. The reviewers also acknowledge our soundness and contribution as seen in the comments of reviewer AwiV who writes *“The paper presents several technical contributions, which overall leads to a more generalizable framework for inertial odometry. Mixing the physical properties with a learning-based regression module makes sense, which can boost generalization performance”* . This sentiment is mirrored by Reviewer EZfo who says *“The combination of technical insights and comprehensive empirical validation underlines the paper's contribution to advancing neural inertial odometry, particularly in settings with challenging orientations and device constraints”*. The Reviewer wVti appreciates the clarity of our paper in their comments by stating the paper is *“written well with clear figures and many intuitive explanations of complex concepts”*. Reviewer EZfo also writes *“Clarity is maintained through structured explanations of complex mathematical formulations, specifically around group theory and its relevance to sensor data processing”*. Reviewer cuLt writes *“The symmetry of the problem in terms of O_g(3) equivariance (which is O(3) subequivariant) is well presented”*.

---

### Meta-Review · Area_Chair_rGgf · 2024-12-20

**Metareview:**

This paper tackles neural inertial odometry estimation, and seeks to regularize training by developing a network that is equivariant to rotations over x,y in a gravity-aligned frame. Theoretically, this is a more principle approach that the earlier methods of regularization via data augmentation and the design of the subequivariant layers is a neat contribution. Empirically, this leads to improvement over baselines and the contributions/design choices are well-ablated method. The reviewers were generally positive about this work, and the AC concurs with this and recommends acceptance. The authors are encouraged to make the text a bit more accessible in the final version based on the reviewer comments.

**Additional Comments On Reviewer Discussion:**

R-wVti had clarity related concerns regarding how the method differs from RIO, and the AC believes the author response adequately clarified this (although R-wVti  did not update their final rating).

---

### Decision · Program_Chairs · 2025-01-22

Accept (Poster)